# Metabolomic Analysis of Goji Berry Sun-Drying: Dynamic Changes in Small-Molecule Substances

**DOI:** 10.3390/foods14244241

**Published:** 2025-12-10

**Authors:** Yao Zhang, Hui Ma, Wan-Ting Zou, Yan-Yan Zhu, Gui-Juan Ma, Yi Lv, Yong-Jie Yu

**Affiliations:** 1Key Laboratory of Quality and Safety of Wolfberry and Wine for State Administration for Market Regulation, Ningxia Food Testing and Research Institute, Yinchuan 750002, China; znxsjy@163.com (Y.Z.); 18195171697@163.com (Y.-Y.Z.); 15296976688@163.com (G.-J.M.); 2 College of Pharmacy, Ningxia Medical University, Yinchuan 750004, China; mah1100@126.com (H.M.); 13469614954@163.com (W.-T.Z.); 3College of Animal Science and Technology, Ningxia University, Yinchuan 750002, China

**Keywords:** goji berries, quality markers, chemometrics, metabolite profiling

## Abstract

Goji berries (*Lycium barbarum* L.) are valued for their nutritional and medicinal properties; however, the systematic biochemical impact of drying on their quality remains poorly understood. This study applied an untargeted metabolomics approach based on UHPLC-HRMS and AntDAS to profile metabolic changes during sun-drying. Multivariate analyses (PCA and PLS-DA) revealed distinct time-dependent clustering, indicating significant shifts in the metabolome. Key metabolites, including betaine, galactose, and *trans*-ferulic acid, increased significantly (*p* < 0.05), whereas choline, allantoin, and huperzine isomers decreased. Pathway analysis highlighted glycine, serine, threonine, galactose, and phenylpropanoid metabolism as the central pathways that were affected. These differential metabolites could potentially be used as quality biomarkers. Our findings establish untargeted metabolomics as an effective tool for elucidating the evolution of goji berry quality during drying, offering a theoretical basis for process optimization.

## 1. Introduction

Goji berries (*Lycium barbarum* L.) represent a distinctive agricultural product of significant economic value [1,2], with wide distribution across Asia [3,4], the Americas, and Africa [5]. Having gained global recognition in the functional food market, their rich nutritional composition and diverse bioactive compounds have garnered substantial attention from the scientific community in recent years. As a high-value agricultural commodity, the processing quality of goji berries directly dictates their market competitiveness and industrial profitability [6]. Goji berries are particularly rich in a suite of bioactive compounds, which are primarily responsible for their renowned health-promoting effects. These include *Lycium barbarum* polysaccharides (LBPs), which have demonstrated immunomodulatory and antioxidant activities; zeaxanthin dipalmitate, which supports eye health by protecting against blue light; and a spectrum of phenolics that contribute to their free radical scavenging capacity [7].

Drying is a critical processing step in the post-harvest chain that extends the shelf life of products and facilitates their transportation and storage [8]. Among the various dehydration techniques, sun-drying remains a widely used traditional method owing to its cost-effectiveness and operational simplicity. However, this process constitutes a form of processing-induced abiotic stress that profoundly influences sensory quality, retention of bioactive compounds, and the final market value of the product [9]. Therefore, an optimized drying process must not only inhibit microbial activity but also preserve the unique flavor and nutritional constituents of the berries to the greatest extent possible [10]. Dehydration stress triggers a cascade of physiological and biochemical responses, including alterations in primary and secondary metabolites, which ultimately determine the quality of the final dried product.

Traditionally, the quality evaluation of goji berries during drying has focused on a limited set of parameters. These studies have primarily quantified changes in key bioactive components, such as polysaccharides and carotenoids [11,12], or monitored physical attributes, such as color and hardness [13]. Although informative, these targeted approaches provide a fragmented view of the complex biochemical transformations that occur within fruit matrices. Consequently, a comprehensive understanding of dynamic shifts in the overall metabolome during drying remains largely uncharacterized.

Metabolomics, an analytical paradigm for the systemic study of small-molecule metabolites, offers distinct advantages for investigating post-harvest physiology and food processing [14]. Sena Bakir et al. utilized an untargeted metabolomics approach to reveal significant alterations in lycopene and vitamin content in response to different drying methods [15]. Using HS-SPME-GC-MS, Aulia demonstrated that pre-drying treatment effectively reduced bitterness and astringency by modulating the flavor profiles of cocoa beans [16]. In a separate investigation, Yu et al. systematically tracked dynamic changes in proteins and amino acids during postharvest tea processing [17]. Although some studies on goji berries have employed targeted metabolomics to examine specific classes of compounds, such as flavonoids and alkaloids [18,19], the global metabolic evolution during drying has not been fully elucidated. The untargeted approach of metabolic fingerprinting is particularly powerful for discovery-based studies. However, it generates vast and complex datasets that present significant analytical challenges, including high false-positive rates and low confidence in compound identification.

To overcome these analytical hurdles, this study introduces a novel method for pesticide residue compound identification that combines the advanced automated data analysis software AntDAS V1.0 with multi-fragment ion analysis. The AntDAS platform demonstrated distinct advantages in the real-time monitoring of metabolic changes during natural sun-drying processes. This integrated system enhances the accuracy of compound identification, improves anti-interference capabilities, and optimizes data utility for complex samples, thereby serving as a powerful tool for identifying novel potential quality markers and facilitating the development of enhanced quality control standards. This methodology has been successfully applied for metabolite screening of *Elaeagnus angustifolia* L. and *Lonicera japonica* Thunb [20,21] and for the identification of adulteration in flaxseed oil [22], demonstrating its practical utility in non-targeted analytical systems.

Building on this validated AntDAS-based non-targeted metabolomics platform, this study aimed to decipher the metabolic evolution and systemic biochemical mechanisms of goji berries during traditional sun-drying. The primary objectives were to identify the key metabolic pathways affected by the dehydration process and discover potential biomarkers linked to the development of color, flavor, and texture, thereby establishing a scientific foundation for process optimization and quality control.

## 2. Materials and Methods

### 2.1. Chemicals and Reagents

High-performance liquid chromatography (HPLC) grade methanol, acetonitrile, dichloromethane, and isopropanol were purchased from Fisher Scientific (Pittsburgh, PA, USA). Liquid chromatography-mass spectrometry (LC-MS)-grade formic acid (98–100%) was purchased from Merck (Darmstadt, Germany). Distilled water was obtained from Watsons Water Company, Inc. (Guangzhou, China). Dispersive solid-phase extraction (d-SPE) extraction kits and purification tubes were acquired from ANPEL Laboratory Technologies, Inc. (Shanghai, China). Analytical standards for imidacloprid, thiamethoxam, difenoconazole, pyraclostrobin, pyridaben, carbendazim, etoxazole, matrine, spinosad, and acetamiprid were procured from Alta Scientific Co., Ltd. (Chengdu, China).

### 2.2. Plant Material and Drying Method

Fresh goji berry samples were collected from Zhongning County, Ningxia, China (latitude 37.52° N, Longitude 105.68° E, Altitude ~1200 m). The goji berry cultivar was Ningqi No. 10, derived from trees for 4 years, and the harvest occurred on 12 July 2024. Fresh goji berry samples were subjected to conventional Na_2_CO_3_ dewaxing treatment [23,24]. Following washing, the goji berries were spread evenly on bamboo mats and placed under direct sunlight for dehydration. The drying process was initiated with an initial sampling at 2 h, followed by daily sampling at the same time until complete desiccation was achieved. Throughout this period, the ambient conditions were maintained at 31 ± 2 °C and approximately 40% relative humidity. For each drying time point, data was derived from ten (n = 10) independent biological replicates. Each replicate originated from a distinct batch of raw materials that were processed separately throughout the experiment. Figure 1 shows the goji berries during sun-drying.

### 2.3. Compound Extraction and Instrument Analysis

Extraction was performed using a modified QuEChERS method [25]. Goji berry powder (1.0 g) was weighed into a 50 mL plastic centrifuge tube. Subsequently, deionized water (9 mL) was added and the mixture was vortex-mixed and maintained overnight at room temperature. Acetonitrile (10 mL) and a ceramic homogenizer were added, and the mixture was agitated vigorously for 1 min. An extraction salt mixture comprising 6 g of anhydrous magnesium sulfate and 1.5 g of sodium acetate was added immediately and shaken to ensure dispersion. The mixture was then homogenized using an automatic vortex mixer for 1 min and centrifuged at 4200× *g* for 5 min. An aliquot of the supernatant (5 mL) was transferred to a d-SPE purification tube containing 150 mg magnesium sulfate, 50 mg primary secondary amine (PSA), 50 mg C18, and 25 mg graphitized carbon black (GCB). The mixture was vortexed for 1 min and centrifuged at 4200× *g* for 5 min. The resulting supernatant was filtered through a 0.22 μm PTFE syringe filter prior to instrumental analysis.

### 2.4. Instrumental Optimization for UHPLC-HRMS Analysis

The goji berry sample extracts were analyzed using an AB SCIEX TripleTOF™ 5600 Plus mass spectrometer (SCIEX, Framingham, MA, USA) for data acquisition. The UHPLC system was operated and data were acquired using Analyst^®^ software (version 1.7.1). The samples were analyzed by sequential injection, in which a quality control (QC) sample was injected every 10 samples throughout the analytical sequence. Chromatographic separation of the compounds was achieved using a Waters ACQUITY UPLC HSS T3 column (2.1 × 100 mm, 1.8 µm) maintained at 40 °C. The mobile phase consisted of (A) an aqueous solution of 0.1% formic acid and (B) an acetonitrile solution of 0.1% formic acid. A flow rate of 0.2 mL/min was used, and the elution gradient was optimized as follows: 0 min, 97% A; 1.5 min, 85% A; 23 min, 50% A; 28 min, 30% A; 33 min, 2% A; 37 min, 97% A; 40 min, 97% A. Electrospray ionization was employed for data acquisition, with the data-dependent acquisition (DDA) mode selected for data collection.

Regarding the mass spectrometry (MS) parameters, the gas pressures for GS1 and GS2 were set to 50 psi. The curtain gas pressure was set to 30 psi, and the ion-source temperature was maintained at 500 °C. The Ion Source Voltage Factor (ISVF) and Declustering Potential (DP) were set at 5500 and 80 V, respectively. The DDA mode was operated with an MS1 scanning range of 100–1000 Da at a resolution of 15,000. The scanning time for each MS1 spectrum was 200 ms. The TopN parameter was set to five ions, which were automatically selected for MS/MS spectrum collection with a precursor ion mass tolerance of 100 mDa. The MS/MS scanning range was set to 50–1000 Da, and the scanning time for each MS/MS spectrum was 30 ms. Collision Energy (CE) to produce ions was 35 V, with a Collision Energy Spread (CES) of 15 V. A dynamic exclusion time of 5 s was used [26].

### 2.5. Data Analysis by Chemometrics

In this study, an in-source fragment ion-based pesticide screening and identification (ISF-PSI) strategy was developed that exploits multiple in-source fragment ions that are naturally generated during the ionization process in high-resolution mass spectrometry. The data analysis workflow in AntDAS-DDA comprises several key steps. Initially, chromatograms were generated by arranging the acquired ions into extracted ion chromatograms (EICs). Subsequently, feature extraction was performed to detect the chromatographic peaks within each EIC. Following this procedure, MS/MS spectra were constructed for the identified features, and fragment ions were subsequently identified to link them to their corresponding compounds.

The acquired UHPLC-HRMS data were processed for compound extraction using AntDAS (http://www.pmdb.org.cn/antdas2, accessed on 1 August 2024). After processing with AntDAS-DDA, the sample data yielded a compound information list based on MS1 chromatographic peaks across different samples. This list was subsequently used for chemometric analyses, including Analysis of Variance (ANOVA), partial least-squares discriminant analysis (PLS-DA), and Principal Component Analysis (PCA). Finally, the mass spectra of the compounds were matched and identified using a third-party database. The differential metabolites identified during the drying process were visualized using a heatmap.

ANOVA was employed to screen compounds that exhibited significant differences in drying time. Chemometric methods, including orthogonal partial least squares discriminant analysis (OPLS-DA), were performed using SIMCA14.1 (Umetrics, Umeå, Sweden). Heatmap analysis was performed using MATLAB 2018b.

## 3. Results and Discussion

### 3.1. Compound Feature Extraction and Multivariate Statistical Analysis

Figure 2 illustrates the Total Ion Chromatogram (TIC) of representative goji berry samples obtained during the drying process. The TIC contained numerous chromatographic peaks, and the TIC of the goji berry samples underwent significant modifications as the drying progressed. The acquired UHPLC-HRMS data were imported into AntDAS for analysis, yielding a feature list with dimensions of 9566 × 80, where 9566 denotes the number of features and 80 represents the number of samples. Features that exhibited significant differences across different drying times were selected by establishing a *p*-value threshold of 0.05 in the AntDAS. These selected features were subsequently grouped into various compounds to construct MS1 and MS2 spectra for compound identification. Significantly different features were identified via ANOVA, followed by multiple comparison correction using the Benjamini–Hochberg False Discovery Rate (FDR) method. Features with an FDR-adjusted *p*-value (q-value) of less than 0.05 were considered significant, resulting in the selection of approximately 3969 features. The samples were clustered based on the screened features, and the corresponding PCA and PLS-DA results are shown in Figure 3. The robustness of the supervised PLS-DA model was confirmed using validation parameters (R^2^Y = 0.9991, Q^2^ = 0.9889 and a significant permutation test (*p* < 0.05), indicating that the model was valid and not overfitted. The temporal evolution of the metabolome of goji berries during sun-drying was investigated using unsupervised and supervised multivariate analyses. Principal Component Analysis (PCA) revealed a distinct trajectory of metabolic profiles along principal components 1 and 2, where samples progressively shifted from day 1 to day 7 (Figure 3a). This continuous drift indicates a systematic and time-dependent transformation of metabolite composition throughout the drying process. To further maximize the separation between drying days, partial least squares discriminant analysis (PLS-DA) was performed. The PLS-DA score plot demonstrated tight day-specific clustering with clear segregation among the early, middle, and late stages of drying (Figure 3b). The robust discrimination achieved by the PLS-DA model confirmed that sun-drying duration is a major driver of profound metabolic changes. Collectively, these analyses underscore that sun-drying induces dynamic and significant alterations in the metabolome of goji berries.

### 3.2. Screening of Differential Compounds and Their Content Distribution

The MF for each compound was calculated as the cosine value of the mass spectrum derived from AntDAS-LCHRMS and that present in the library. Compound identification was performed using AntDAS-LCHRMS, and the results were manually verified using a graphical user interface (GUI) for compound identification. A total of 27 compounds were identified, comprising 18 Active Ingredients, 4 pesticides, and 5 others. Detailed information on the identified compounds is provided in Table 1.

### 3.3. Changes in Key Metabolites During the Natural Drying Process of Goji Berries

Goji berries, recognized for their medicinal and nutritional properties, are abundant in carbohydrates, amino acids, carotenoids, polyphenols, and secondary metabolites in their fresh state. During drying, various compounds undergo systematic degradation and transformation, which are influenced by factors such as water evaporation, endogenous enzyme activity, and microbial action. These conversions proceed via metabolic pathways, such as the sugar, amino acid, phenylpropanoid, and carotenoid metabolism pathways. These transformations ultimately affect the sensory quality and nutritional properties of dried fruits.

A total of 27 metabolites were identified as being significantly altered during the drying process. It should be noted that these identifications were assigned with a confidence level of 2 according to the Metabolomics Standards Initiative (MSI), as they were putatively annotated based on accurate mass and MS/MS spectral matching using the AntDAS platform, without validation with authentic standards. Several metabolites, including betaine, galactose, and *trans-ferulic acid*, showed increased concentrations as the drying time progressed. Of the 27 metabolites analyzed, 16 exhibited an upward trend during drying, whereas the remaining 12 showed a decrease. Notably, metabolites such as betaine [27], phenylalanine [28], and *trans-ferulic acid* [29] play pivotal roles in determining the flavor and sensory attributes of goji berries. The relative concentrations of betaine, galactose, and *trans-ferulic acid* increased with prolonged drying time, whereas those of choline, uracil, and huperzine A decreased (Figure 4).

#### 3.3.1. The Impact of Drying Processes on Carbohydrates

The sensory attributes of goji berries are critically dependent on the concentrations of their constituent sugars and amino acids. In fresh goji berries, polysaccharides are catabolized into monosaccharides and oligosaccharides via glycosidase enzymatic activity. This study demonstrated that the relative abundance of galactose and betaine increased rapidly during the initial phases of drying before reaching a plateau (Figure 4). Galactose, a principal contributor to sweetness, exhibits considerable thermal stability during drying and is a reactive substrate for the Maillard reaction. This observation corroborates previous findings, where non-enzymatic browning reactions during drying induced color alterations in both the fruit flesh and skin [30]. Congruent transformation patterns were observed for other saccharides during the drying period. Collectively, these data suggest that polysaccharide catabolism in goji berries is most pronounced during the early stages of drying. This is consistent with the findings of López et al., who reported a significant reduction in sucrose content concurrent with an increase in β-carotene content as the drying temperature increased in berries [31].

As the drying process progresses, sugars are catabolized through cellular respiration to fuel metabolic processes. A portion of these sugars is metabolized into intermediates, such as pyruvate, via the glycolytic pathway, thereby supplying energy for other essential cellular functions. Metabolic activity results in a progressive reduction in glucose, galactose, and trehalose concentrations. A similar trend was documented for durians during drying, where the total sugar content initially decreased at a rapid rate and subsequently decelerated [32]. The comprehensive metabolic flux of all monitored sugars during drying indicated that carbohydrate metabolism is a pivotal factor influencing the final flavor, texture, and color of dried goji berries. In a related study, Bi et al. established a correlation between the levels of 3-deoxyglucosone and 1-deoxy-2,3-pentose ketose and the color changes in rape bee pollen during hot-air drying [33]. Furthermore, sugars function as critical osmolytes for maintaining cellular water retention. Their depletion disrupts the osmotic potential of cells and accelerates water loss. This phenomenon drives the textural transition from a plump and tender state to a shrunken and desiccated state. This mechanism aligns with the findings of Lahaye in tomatoes, where the content of pectin-associated sugars, mannose, and glucose decreased significantly during drying, leading to reduced hydration around pectin molecules and increased intermolecular interactions within the cell wall matrix [34].

#### 3.3.2. Changes in Amino Acids and Phenolic Compounds

The interplay between amino acid and phenolic pathways critically influences the ultimate flavor and color profiles of goji berries. Hou et al. observed that the browning of persimmons during drying was primarily attributable to the oxidation of phenolic compounds, accompanied by an increase in phenylalanine content after drying [35]. Similarly, Liu et al. discovered that phenylalanine could upregulate reactive oxygen species (ROS) metabolism and the phenylpropanoid pathway, thereby enhancing spoilage control in citrus fruits [36]. In this study, phenylalanine, the primary substrate of the phenylpropanoid pathway, showed a continuous increase throughout the drying period. It is progressively converted via enzymatic reactions into downstream metabolites, including *trans-4-coumaric acid*, rosmarinic acid, and other polyphenolic and coumarin-like compounds. Over 1–7 days, the trans-ferulic acid content increased to 3.5 times its initial level (Figure 4). These results are consistent with those of a study by Wang et al., who reported that melatonin treatment of goji berries elevated the levels of coumaric and ferulic acids, which subsequently activated the antioxidant systems and secondary metabolism [37]. These phenolic acids play a dual role in determining the quality. They act as substrates for enzymatic browning via oxidation, forming dark brown polymers that alter color [38]. They also contribute to the flavor profile, particularly when they interact with other phenolic aldehydes, such as phenylacetaldehyde. The concentration of phenylacetaldehyde peaked during the intermediate phase of drying and then declined slightly, a pattern likely attributable to the stage-specific kinetics of the Maillard reaction. The accumulation of trans-ferulic acid imparts a characteristic herbal aroma and enhances the overall antioxidant capacity of the final product [39].

Umami-associated amino acids, specifically glutamic acid and aspartic acid, remained relatively stable during the initial drying phase (1~3 d), an effect likely due to the concentration of solutes as water was removed. However, in the later stages (4~7 d), these amino acids were partially deaminated to form α-ketoglutarate, which subsequently entered the tricarboxylic acid cycle as an anaplerotic intermediate for energy production, leading to a decline in their concentrations. In parallel, Zhang et al. identified the tricarboxylic acid cycle, along with glycine, serine, and threonine metabolism, as the principal metabolic pathways governing peanut drying [40]. Huang et al. also observed a decrease in the amino acid content of goji berries following hot-air drying [41]. In contrast, the levels of betaine, an important antioxidant compound, increased throughout the drying process. Therefore, monitoring the dynamic changes in key metabolites during goji berry drying is an effective method for assessing the progression of quality transformation.

#### 3.3.3. Changes in Alkaloids and Nucleotides

Alkaloids and nucleotides in goji berries significantly contribute to their bioactive properties and potential health benefits. This investigation revealed that several bioactive compounds, including huperzine A, uracil, and adenine, underwent significant degradation during drying (Figure 4). Huperzine A, a functionally important alkaloid, demonstrated marked thermal lability, with its concentration declining sharply after 7 days of drying. This finding indicates that the drying process degraded certain thermolabile bioactive components. Concurrently, the content of uracil, an intermediate in purine metabolism that facilitates nitrogen mobilization and enhances stress tolerance in goji berries, also decreased [40]. These observed changes in nucleotide metabolism align with the findings of Stasolla et al. on purine and pyrimidine metabolic shifts during the drying process of Picea glauca [42].

#### 3.3.4. Effects of Moisture Loss During Drying on the Appearance of Goji Berries

The solar drying process subjects goji berries to synergistic dehydration and thermal stress, which orchestrate coherent biochemical restructuring. Our study demonstrates that temporal changes in key metabolites are interconnected responses that collectively determine the final quality, revealing a central narrative of cellular adaptation at a cost. This narrative is exemplified by the pivotal role of choline metabolism. Its initial stability preserves membrane integrity and texture. However, as drying intensifies, the active depletion of choline for phosphatidylcholine synthesis becomes a hallmark of membrane lipid remodeling to combat dehydration damage [43,44]. This critical adaptation paradoxically increases membrane permeability and creates a vulnerable intracellular environment. Within the context of compromised membrane integrity and sustained heat, the significant decline in thermolabile alkaloids, such as huperzine A, must be viewed. Thus, the loss of huperzine A is not merely a thermal event but is mechanistically linked to the cell’s adaptive response, illustrating a cascade from primary stress to metabolite degradation.

Simultaneously, the structural integrity of the cell wall was systematically deconstructed. Enzymatic depolymerization of pectin into galacturonic acid weakens the intercellular matrix [45,46]. This process is synergistically accelerated by overall water loss, a condition exacerbated by increased membrane permeability due to choline remodeling [47]. This results in a macroscopic transition to a shrunken softened product. Ultimately, the final sensory quality, developed through the Maillard and enzymatic browning pathways [48,49], is tied to the precursor concentrations and thermal history established by earlier metabolic events.

### 3.4. Potential Metabolic Processes During Drying

To further elucidate the metabolic changes occurring during sun-drying of fresh goji berries (1–7 d), a metabolic pathway enrichment analysis was conducted. This analysis identified sugar, amino acids, phenylpropanoids, and nucleotide metabolism as the principal pathways involved in the degradation and transformation of endogenous sugars, amino acids, phenolic compounds, and nucleotides in goji berries.

Systematic mapping of key metabolic processes revealed a highly interconnected network. Sugars undergo glycolysis and gluconeogenesis, yielding critical intermediates such as pyruvate and phosphoenolpyruvate (PEP). Amino acids are channeled into pathways, including the phenylpropanoid biosynthesis route, which generates phenolic compounds, such as *trans-ferulic acid*, and various aromatic constituents, such as phenylacetaldehyde. Nucleotide metabolism regulates material transport and energy transduction processes.

These disparate pathways converge at central metabolic hubs, including the tricarboxylic acid cycle, allowing the interconversion of intermediates. This network facilitates the progressive catabolism of complex biomolecules into smaller constituents, a process accompanied by the continuous release of energy. Consequently, despite favorable drying conditions, the nutrient profile of goji berries undergoes significant degradation and metabolic transformation over time, driven by endogenous biochemical and enzymatic reactions that ultimately affect the sensory and nutritional qualities of the final dried product (Figure 5).

## 4. Conclusions

This study utilized an untargeted metabolomics platform based on Ultra-Performance Liquid Chromatography Quadrupole Time-of-Flight Mass Spectrometry (UPLC-QTOF-MS) coupled with AntDAS data analysis to systematically investigate dynamic metabolic changes during the natural sun-drying of wolfberry. The analysis successfully identified 27 key differential metabolites whose concentrations changed significantly during the drying period. The concentrations of 16 compounds, including betaine, galactose, and *trans-ferulic acid*, increased with drying duration, which correlated directly with the development of sweet and umami taste profiles, flux through the phenylpropanoid pathway, and progression of enzymatic browning. Conversely, the levels of 12 other metabolites, notably choline, allantoin, and huperzine A, decreased, indicating the degradation of thermolabile and bioactive constituents. Metabolic pathway analysis confirmed that the observed transformations in flavor, color, and texture were governed by the collective modulation of sugar, amino acid, phenylpropanoid, and nucleotide metabolism pathways. This study demonstrated the efficacy of the AntDAS-LCHRMS-assisted UPLC-QTOF-MS platform as a powerful tool for elucidating the complex biochemical evolution of quality attributes during food desiccation. These findings provide a robust theoretical foundation for optimizing drying protocols and enhancing quality control measures for commercial wolfberry production.

## Figures and Tables

**Figure 1 foods-14-04241-f001:**
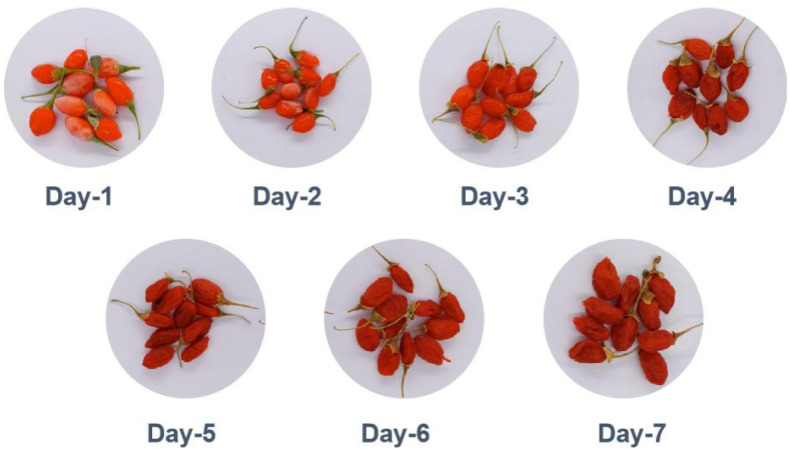
Time course of visual morphological changes in sun-dried goji berries. Images show representative berries at daily intervals (days 1–7) throughout the drying process.

**Figure 2 foods-14-04241-f002:**
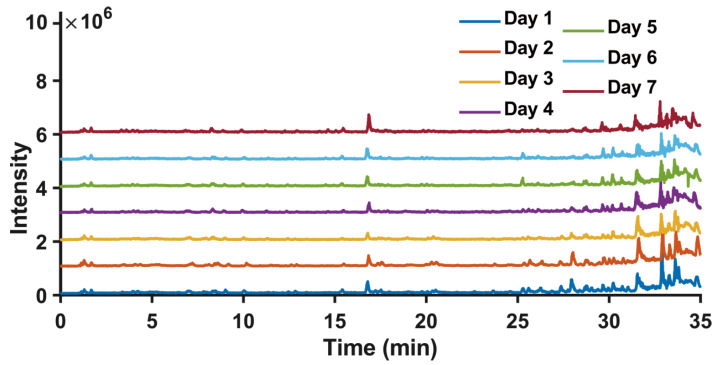
Changes in total ion chromatograms of goji berries over seven days of sun drying. The curves correspond to the samples analyzed after each day of drying (days 1–7).

**Figure 3 foods-14-04241-f003:**
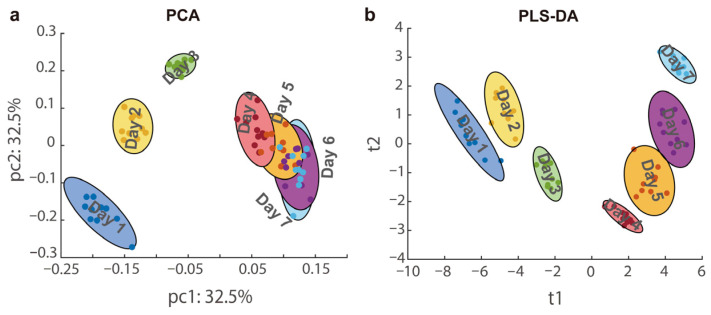
Classification results of PCA and PLS-DA based on the screened features. (**a**) Classification results by PCA; (**b**) classification results using PLS-DA.

**Figure 4 foods-14-04241-f004:**
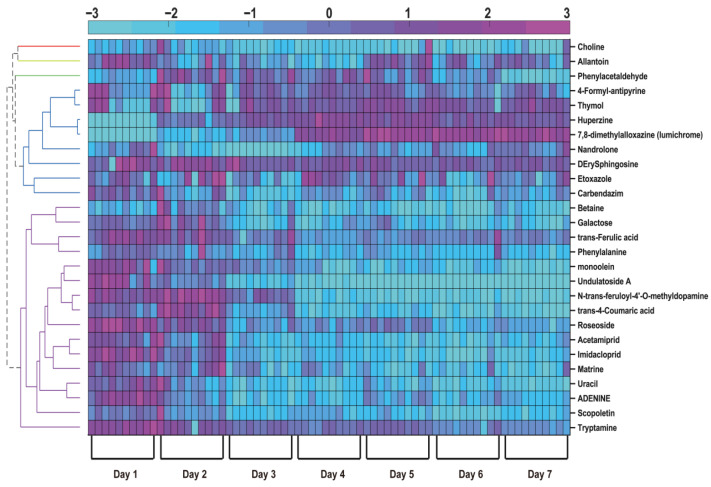
Hierarchical clustering heatmap of 27 differential metabolites in goji berries during sun-drying. Samples were grouped by drying day (1–7). Metabolite levels were scaled as Z-scores (red—high; blue—low).

**Figure 5 foods-14-04241-f005:**
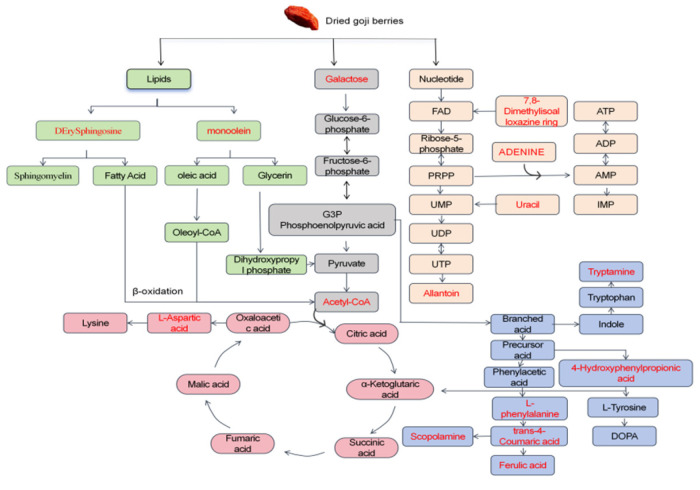
Overview of metabolic pathways in goji berries with highlighted drying-responsive metabolites. The metabolites highlighted in red represent significantly altered differential metabolites identified through statistical screening.

**Table 1 foods-14-04241-t001:** Compound identification results of characteristic markers of *Goji berries* during sun drying. The number ‘1’ in MS1 denotes the first, full-scan stage of mass spectrometry, which detects intact precursor ions, in contrast to the subsequent fragmentation stage (MS/MS).

Num	*m*/*z*	RT (Min)	MF (MS ^1^)	MF (MS/MS)	Name	Formular	Class
1	104.107	1.214	0.999	0.913	Choline	C_5_H_14_NO	Active Ingredients
2	198.097	1.276	0.710	0.342	Galactose	C_6_H_12_O_6_	Active Ingredients
3	118.086	1.291	1.000	0.995	Betaine	C_5_H_11_NO_2_	Active Ingredients
4	159.051	1.377	0.826	0.973	Allantoin	C_4_H_6_N_4_O_3_	Active Ingredients
5	136.061	1.668	1.000	0.945	ADENINE	C_5_H_5_N_5_	Active Ingredients
6	113.034	1.668	0.999	0.794	Uracil	C_4_H_4_N_2_O_2_	Active Ingredients
7	121.964	1.676	0.691	0.584	Phenylacetaldehyde	C_8_H_8_O	Active Ingredients
8	249.196	1.692	0.975	0.995	Matrine	C_15_H_24_N_2_O	Active Ingredients
9	166.086	3.552	0.978	0.909	Phenylalanine	C_9_H_11_NO_2_	Active Ingredients
10	192.077	4.230	0.466	0.984	Carbendazim	C_9_H_9_N_3_O_2_	Pesticides
11	355.101	4.658	0.848	0.995	Undulatoside A	C_16_H_18_O_9_	Active Ingredients
12	387.200	4.875	0.645	0.898	Roseoside	C_19_H_30_O_8_	Active Ingredients
13	161.107	4.947	0.985	1.000	Tryptamine	C_10_H_12_N_2_	Active Ingredients
14	165.054	5.004	0.595	0.988	*trans-4-Coumaric acid*	C_9_H_8_O_3_	Active Ingredients
15	344.148	7.206	0.996	0.977	*N- trans-feruloyl-4′-O-methyldopamine*	C_19_H_21_NO_5_	Active Ingredients
16	173.080	8.285	0.962	0.788	Thymol	C_10_H_14_O	Active Ingredients
17	217.107	8.285	0.946	0.315	4-Formyl-antipyrine	C_12_H_12_N_2_O_2_	Others
18	193.049	8.437	0.956	0.987	Scopoletin	C_10_H_8_O_4_	Active Ingredients
19	195.064	8.573	0.941	0.853	*trans-Ferulic acid*	C_10_H_10_O_4_	Active Ingredients
20	256.059	8.998	0.915	0.989	Imidacloprid	C_9_H_10_ClN_5_O_2_	Pesticides
21	243.087	9.768	0.993	0.977	7,8-dimethylalloxazine	C_12_H_10_N_4_O_2_	Others
22	223.075	10.033	0.907	0.990	Acetamiprid	C_10_H_11_ClN_4_	Pesticides
23	275.200	15.381	0.579	0.666	Nandrolone	C_18_H_26_O_2_	Others
24	360.176	27.365	0.989	0.987	Etoxazole	C_21_H_23_F_2_NO_2_	Pesticides
25	300.289	29.589	0.999	0.983	*D-erythro*-Sphingosine	C_18_H_37_NO_2_	Others
26	243.150	33.095	0.971	0.993	Huperzine	C_15_H_18_N_2_O	Active Ingredients
27	357.299	34.795	0.543	0.850	monoolein	C_21_H_40_O_4_	Others

## Data Availability

The original contributions presented in the study are included in the article; further inquiries can be directed to the corresponding authors.

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
