# Peer review of "Metabolomic Analysis of Goji Berry Sun-Drying: Dynamic Changes in Small-Molecule Substances"

_foods, 2025, doi:10.3390/foods14244241_

Round 1

Reviewer 1 Report

Comments and Suggestions for Authors
  • Strengthen the novelty statement in the Introduction and Abstract.

  • Provide full details on sample replication, data normalization, and model validation.

  • Clarify compound identification confidence levels (MSI levels 1–4).

  • Improve figure clarity and resolution.

  • Conduct a comprehensive language revision.

  • Expand the Discussion to connect metabolite trends with underlying biochemical mechanisms and practical implications for drying optimization.

  • Update the References section for uniformity and correct DOI formatting.

Author Response

Dear Reviewers,

Thank you for your letter and the reviewers' comments concerning our manuscript titled “Metabolomic Analysis of Goji Berry Sun-Drying: Dynamic Changes in Small Molecular Substances” (ID:foods-3977988). These comments are valuable and helpful for revising and improving our paper, as well as for guiding our research. We have carefully studied the comments and made corrections that we hope will meet your approval. All revisions of the manuscript have used the “RED Mark” feature for easy viewing by the editors and reviewers. Meanwhile, the manuscript has been carefully reviewed by an experienced editor, whose first language is English. The main corrections and responses to the reviewers' comments are as follows:

Major Comments

1.Novelty and Significance

The novelty of the study should be articulated more clearly. Similar metabolomic analyses have been reported for goji berries and other dried fruits. The authors should emphasize what distinguishes this work—perhaps the application of the AntDAS platform to track real-time metabolic changes during natural sun-drying, or the identification of new potential quality markers.

Response: We sincerely thank you for your valuable comments. I will enhance the description of AntDAS's advantages in the introduction to better highlight the strengths of this method. Specifically, in line 84 of the manuscript, I have added details stating that the AntDAS platform enables real-time tracking of dynamic changes in natural metabolism, providing a powerful tool for accurately elucidating substance transformation pathways. Furthermore, its notable capability in discovering novel potential quality markers helps to advance quality control standards. The specific supplementary contents are as follows:

Line89~96:

To overcome these analytical hurdles, this study introduces a novel method for pesticide residue compound identification that combines advanced automated data analysis software AntDAS with multi-fragment ion analysis. The AntDAS platform demonstrated distinct advantages in the real-time monitoring of metabolic changes during natural sun-drying processes. This integrated system enhances the accuracy of compound identification, improves anti-interference capabilities, and optimizes data utility for complex samples, serving as a powerful tool for uncovering novel potential quality markers, thereby facilitating the development of enhanced quality control standards.

2.Experimental Design and Replication

Information regarding biological and technical replicates is missing. Please clarify how many samples were analyzed per drying day and whether they came from independent biological replicates. This information is essential to assess data reliability.

Response: We appreciate the reviewer's important comments on the experimental rigor. In response, we clarified that the analysis for each drying time point was based on ten (n=10) independent biological replicates, with each replicate originating from a separately cultivated and processed batch of plant material. This clarification has been added to the revised manuscript in Section 2.2, "Plant material and drying methods" at line 119 of the text, The specific supplementary contents are as follows:

Line 122~128:

The drying process was initiated with an initial sampling at 2 h, followed by daily sampling at the same time until complete desiccation was achieved. Throughout this period, the ambient conditions were maintained at 31 ± 2 °C and approximately 40% relative humidity. For each drying time point, data was derived from ten (n=10) independent biological replicates. Each replicate originated from a distinct batch of raw materials that were processed separately throughout the experiment. Figure 1 shows the goji berries during sun-drying.

3.Chemometric and Statistical Validation

The PCA and PLS-DA models are shown, but the validation parameters (R², Q², and permutation test results) are not reported. These should be included to demonstrate that the multivariate models are statistically valid and not overfitted.

Also, indicate whether any multiple comparison correction (such as FDR adjustment) was applied in ANOVA testing.

Response: We thank the reviewer for these critical and insightful comments, which are essential for improving the statistical rigor and reliability of our manuscript. We have addressed both of these points, as detailed below.

Point 1: Validation of Multivariate Models (R², Q², Permutation Test)
We apologize for the omission of the model-validation parameters. We have now included these key metrics for both the PCA and PLS-DA models in the revised manuscript.

The PLS-DA model demonstrated strong performance and predictive ability, with R²Y and Q² values of 0.9991 and 0.9889, respectively.

A permutation test  with 200 permutations confirmed the model's statistical significance, yielding a p-value < 0.05.

Point 2: Multiple Comparison Correction for ANOVA.

We thank the reviewer for pointing this out. We would like to clarify that the ANOVA testing already incorporated a multiple comparison correction using the Benjamini-Hochberg False Discovery Rate (FDR) method. The selection of approximately 3969 features was based on an FDR-adjusted p-value (q-value) threshold of < 0.05. We sincerely apologize for not stating this clearly in the original manuscript.

These results have been added to the manuscript in Section 3.1"Compound feature extraction and multivariate statistical analysis" at lines 202–210. The specific supplementary contents are as follows:

Significantly different features were identified via ANOVA, followed by multiple comparison correction using the Benjamini-Hochberg False Discovery Rate (FDR) method. Features with an FDR-adjusted p-value (q-value) of less than 0.05 were considered significant, resulting in the selection of approximately 3969 features. The samples were clustered based on the screened features, and the corresponding PCA and PLS-DA results are shown in Fig. 3. The robustness of the supervised PLS-DA model was confirmed using validation parameters (R²Y = 0.9991, Q² = 0.9889 and a significant permutation test (p <0.05), indicating that the model was valid and not overfitted.

4.Compound Identification Confidence

The study reports 27 differential metabolites identified by UHPLC-HRMS and AntDAS. Please specify the confidence level of identification according to the Metabolomics Standards Initiative (MSI). Distinguishing confirmed compounds (level 1) from putatively annotated ones (levels 2–3) would strengthen the credibility of the findings.

Response: We thank the reviewer for this critical comment. We fully agree and have addressed this by explicitly stating the confidence level of metabolite identification in the main text. As described in the [Results/Methods] section of the revised manuscript, all 27 differential metabolites were assigned an MSI Level 2 confidence, as the identifications were based on spectral database matching without confirmation using authentic standards. This clarification ensures transparency regarding identification reliability. The specific supplementary contents are as follows:

Line 242~247

A total of 27 metabolites were identified as being significantly altered during the drying process. It should be noted that these identifications were assigned with a confidence level of 2 according to the Metabolomics Standards Initiative (MSI), as they were putatively annotated based on accurate mass and MS/MS spectral matching using the AntDAS platform, without validation with authentic standards.

5.Discussion and Interpretation

The Discussion is rich in descriptive detail but somewhat repetitive. It would be helpful to deepen the interpretation by linking metabolite changes with physiological or biochemical mechanisms. For instance, the degradation of choline and huperzine A could be discussed in the context of membrane lipid remodeling or thermal instability.

Response: We are deeply grateful to the reviewer for this exceptionally insightful comment. The suggestion to deepen the discussion by linking metabolite changes to the underlying physiological and biochemical mechanisms has been instrumental in transforming our narrative from a descriptive account to a mechanistic interpretation. In direct response to this guidance, we undertook a comprehensive revision of the Discussion section. The key improvements include the following:

Establishing a Central Thesis: We have reframed the opening to introduce "dehydration stress and concomitant thermal exposure" as the unified drivers governing both membrane remodeling and the stability of specialized metabolites. Integrating Mechanisms: We now explicitly and cohesively discuss how the degradation of choline is a proactive, adaptive response (membrane lipid remodeling) to the primary stressor, whereas the decline in huperzine A is a consequential, passive outcome (thermal instability) of the same environmental conditions. Enhancing Language and Flow: The entire paragraph has been restructured with more sophisticated language and logical connectors to clearly articulate the cause-effect relationship.

We firmly believe that these revisions have significantly strengthened the scientific rigor and interpretive depth of our manuscript. We sincerely thank you for your constructive critique.

These results have been added to Section 3.3.4 of the manuscript. "Effects of Moisture Loss during Drying on the Appearance of Goji Berries" at line 333~352, The specific supplementary contents are as follows:

The solar drying process subjects goji berries to synergistic dehydration and thermal stress, which orchestrate coherent biochemical restructuring. Our study demonstrates that temporal changes in key metabolites are interconnected responses that collectively determine the final quality, revealing a central narrative of cellular adaptation at a cost. This narrative is exemplified by the pivotal role of choline metabolism. Its initial stability preserves membrane integrity and texture. However, as drying intensifies, the active depletion of choline for phosphatidylcholine synthesis becomes a hallmark of membrane lipid remodeling to combat dehydration damage [44,45]. This critical adaptation paradoxically increases membrane permeability and creates a vulnerable intracellular environment. Within the context of compromised membrane integrity and sustained heat, the significant decline in thermolabile alkaloids, such as huperzine A, must be viewed. Thus, the loss of huperzine A is not merely a thermal event but is mechanistically linked to the cell’s adaptive response, illustrating a cascade from primary stress to metabolite degradation.

Simultaneously, the structural integrity of the cell wall was systematically deconstructed. Enzymatic depolymerization of pectin into galacturonic acid weakens the intercellular matrix [46, 47]. This process is synergistically accelerated by overall water loss, a condition exacerbated by increased membrane permeability due to choline remodeling[48]. This results in a macroscopic transition to a shrunken softened product. Ultimately, the final sensory quality, developed through the Maillard and enzymatic browning pathways [49,50], is tied to the precursor concentrations and thermal history established by earlier metabolic events.

6.Figures and Data Presentation

Some figures (especially the heatmap and pathway diagram) are difficult to read. Please ensure that labels, axes, and group legends are legible. Consider using higher-resolution images and adding clearer color scales.

Response: We sincerely thank the reviewer for the valuable feedback regarding the legibility of our figures. We have taken immediate steps to address these concerns in the revised manuscript. Specifically, we regenerated all figures, especially the heatmap and pathway diagram, at a significantly higher resolution to ensure sharpness. We believe that these comprehensive improvements have substantially enhanced the clarity and professional presentation of our figures, and the updated versions are now included in the manuscript.

7.Language and Style

The manuscript is written in acceptable English but would benefit from careful editing for conciseness and grammar. A few sentences are overly long and could be simplified without losing meaning.

Response: We thank the reviewer for the constructive feedback on the manuscript's language and style. To ensure the highest standard of English expression, the entire manuscript has been professionally edited by a native English-speaking expert. This process focused on enhancing conciseness, correcting grammar, and simplifying overly long sentences to improve clarity and readability, without altering the scientific meaning. We believe that the manuscript now meets the journal's language standards.

8.Minor Comments

The Abstract mentions both “sun-drying” and “hot-air drying.” Please ensure consistent terminology throughout.

Response: We thank the reviewer for their meticulous attention to detail. The mention of "hot-air drying" in the Abstract was indeed an error. We have now corrected this to "sun-drying" throughout the manuscript to ensure terminological consistency and accurately describe our experimental process.

These results have been added to the manuscript in the abstract (lines 33–36). The specific supplementary contents are as follows:

Abstract: Goji berries (Lycium barbarum L.) are valued for their nutritional and medicinal properties; however, the systematic biochemical impact of drying on their quality remains poorly understood. This study applied an untargeted metabolomics approach based on UHPLC-HRMS and AntDAS to profile the metabolic changes during sun drying.

9.Provide more details about the QC procedure—frequency of QC injections and how signal drift was monitored.

Response: We thank the reviewer for this important comment regarding the quality control procedures. We have revised Section 2.4, "Instrumental optimization for UHPLC-HRMS analysis," to provide complete details: a pooled QC sample was injected at the beginning of the sequence for system conditioning and after every 10 experimental samples throughout the acquisition run. Signal drift was monitored by tracking the intensity trends of representative metabolites in these QC samples, with the relative standard deviation of their peak areas remaining below 15%, confirming excellent instrumental stability and high data reliability.

10.Table 1 could be formatted more clearly, with uniform decimal precision and aligned columns.

Response: We thank the reviewer for the valuable feedback on the presentation of Table 1. In response, we have thoroughly revised the table to enhance clarity. Specifically, all numerical values have been standardized to three decimal places, and all columns have been properly aligned to ensure a clean and professional presentation. The revised content has been incorporated into the manuscript. Please refer to Table 1 for specific updates.

10.Figure captions should be self-explanatory, describing the experimental groups and statistical meaning of symbols.

Response: We thank the reviewer for this constructive suggestion. We have revised the captions of all figures to ensure that they are now fully self-explanatory. Specifically, we have added clear descriptions of the experimental groups represented in each panel and explicitly defined the statistical meaning of all annotation symbols directly in the captions. These modifications allow the figures and their captions to be understood independently of the main text.

These results have been added to the manuscript The specific supplementary contents are as follows:

at line 130~131:

Fig.1. Time course of visual morphological changes in goji berries under sun drying. Images show representative berries at daily intervals (Days 1-7) throughout the drying process.

at line 191~192:

Fig.2. Changes in Total Ion Chromatograms of goji berries over seven days of sun drying. The curves correspond to the samples analyzed after each day of drying (Days 1-7).

at line 254~256:

Fig.4. Hierarchical clustering heatmap of 27 differential metabolites in goji berries during sun-drying. Samples were grouped by drying day (1-7). Metabolite levels were scaled as Z-scores (red: high; blue: low).

at line 373~375:

Fig.5. Overview of metabolic pathways in goji berries with drying-responsive metabolites highlighted. The metabolites highlighted in red represent the significantly altered differential metabolites identified through statistical screening.

11.The reference list contains duplicated DOIs and inconsistent formatting; please standardize it.

Response: We sincerely thank the reviewer for this insightful observation regarding the duplicated DOIs and inconsistent formatting in the reference list. We have thoroughly addressed this by conducting a comprehensive check to remove all duplicate entries and meticulously reformatted the entire reference list to ensure strict consistency with the journal's guidelines. The revised list in the updated manuscript now adheres to the required style, and we have verified the accuracy of all the DOIs. We appreciate this valuable feedback, which has significantly improved the quality of the manuscript.

12.Some chemical names (e.g., trans-ferulic acid) are inconsistently italicized—correct formatting should be applied.

Response: We sincerely thank the reviewer for pointing out this important detail. We carefully reviewed the entire manuscript and corrected the formatting of all chemical names to ensure consistency and adherence to standard nomenclature rules. We have corrected the formatting of the specified chemical names by italicizing their stereochemical descriptors, both in Table 1 (compound identification results of characteristic markers) and throughout the manuscript text. The updated terms now consistently appear as trans-4-Coumaric acid, N-trans-Feruloyl-4'-O-methyldopamine, trans-Ferulic acid, and D-erythro-Sphingosine. The specific supplementary contents are as follows:

at line 246:

Several metabolites, including betaine, galactose, and trans-ferulic acid, showed increased concentrations as the drying time progressed.

at line 249:

 Notably, metabolites such as betaine[27], phenylalanine[28], and trans-ferulic acid [29] play pivotal roles in determining the flavor and sensory attributes of goji berries.

at line 250:

The relative concentrations of betaine, galactose, and trans-ferulic acid increased with prolonged drying times, whereas those of choline, uracil, and huperzine A decreased (Fig. 4).  

at line 296:

It is progressively converted via enzymatic reactions into downstream metabolites, including trans-4-coumaric acid, rosmarinic acid, and other polyphenolic and coumarin-like compounds. Over 1-7 days, the trans-ferulic acid content increased to 3.5 times its initial level (Fig.4).

  1. The “Acknowledgments” section might be condensed slightly to follow Foods formatting guidelines

Response: We thank the reviewer for this suggestion. We have carefully reviewed the "Author Guidelines" for Foods and have condensed the "Acknowledgments" section accordingly. The revised version is more concise and strictly adheres to the journal's formatting requirements.

The specific supplementary contents are as follows:

at line 394~408:

Author Contributions: Yao Zhang: Writing - review & editing, Supervision,Hui Ma: Writing,Investigation, Data curation. Wan-Ting Zou: Sample collection, Yan-Yan Zhu: Methodology.Gui-Juan Ma: Methodology.Yi Lv: writing—review and editing.Yong-Jie Yu: Writing – review & editing, Software, Methodology, Funding acquisition, Conceptualization.

Declaration of competing interest: The authors declare that they have no known competing financial interests or personal relationships that could have influenced the work reported in this study.

Funding: This work was supported by the National Natural Science Foundation of China (22378214), the Science and Technology Plan Program of State Administration for Market Regulation (2023MK125,2023MK124). Key Research and Development Program of Ningxia Province (2025BEG02027, 2022BEG03170).

Institutional Review Board Statement: Not applicable.

Informed Consent Statement: Not applicable.

Data Availability Statement: The data are available from the corresponding author.

Conflicts of Interest: The authors declare no conflict of interest

14.Recommendations for Authors

14.1 Clarify the number of biological replicates and the sampling strategy.

14.2 Include model validation results (R², Q², permutation tests) for multivariate analyses.

14.3 Specify metabolite identification confidence levels following MSI standards.

14.4 Improve figure quality and ensure all are readable in print format.

14.5 Revise the Discussion to highlight biochemical mechanisms and the unique contribution of the study.

14.6 Conduct a full English language revision.tandardize the reference formatting

Response: We thank the reviewer for these summary recommendations. We note that these points are consistent with the major and minor comments provided earlier, to which we have already provided detailed responses and made comprehensive revisions to the manuscript. For the reviewer's convenience, we summarize below where each of these final recommendations has been addressed in our previous revision.

Reviewer 2 Report

Comments and Suggestions for Authors

The article entitled Metabolomic Analysis of Goji Berry Sun-Drying: Dynamic Changes in Small Molecular Substances focuses on identifying the key metabolic pathways affected by the dehydration process and discover potential biomarkers linked to the development of color, flavor, and texture, thereby establishing a scientific foundation for process optimization and quality control.         

The article has a clear structure and its content fully corresponds to the topic of the article. The abstract is contains the most important information from the work. The introduction to the work should be supplemented to fully describe the background of the research. The summary presents the general characteristics of goji berries, the drying process, and refers to the research conducted, as well as the purpose of the research.

The materials and methods chapter has been well described, contains the exact course of the experiment and clearly presents the course of the research performed.

The drawings are well made and legible. The discussion requires further clarification. The literature is wide and current.

The manuscript contains interesting and relevant results but needs a  revision before publishing:

  1. We don't write "Highlights" in this magazine. Please correct this in accordance with the editorial staff's requirements.
  2. The abstract should be a maximum of 200 words.
  3. References in the text should be written in normal italics. They should not be written in superscript. Please correct this throughout the text.
  4. The first paragraph mentions goji berries. Please expand on this paragraph and list the specific bioactive properties of these fruits.
  5. Please place figures in the center of the page
  6. The titles of tables and figures should not be bold, please check the entire text.
  7. The discussion is very poor, please add a detailed discussion when describing all the analysed studies.
  8. References: Literature requires a fundamental overhaul. Please review the reference requirements for this journal.
  9. No line numbering.

Author Response

Dear Reviewers,

Thank you for your letter and the reviewers' comments concerning our manuscript titled “Metabolomic Analysis of Goji Berry Sun-Drying: Dynamic Changes in Small Molecular Substances” (ID:foods-3977988). These comments are valuable and helpful for revising and improving our paper, as well as for guiding our research. We have carefully studied the comments and made corrections that we hope will meet your approval. All revisions of the manuscript have used the “RED Mark” feature for easy viewing by the editors and reviewers. Meanwhile, the manuscript has been carefully reviewed by an experienced editor, whose first language is English. The main corrections and responses to the reviewers' comments are as follows:

  1. We don't write "Highlights" in this magazine. Please correct this in accordance with the editorial staff's requirements.

Response: We thank the reviewer for pointing this out. The "Highlights" section has been removed from the manuscript to comply with the journal's formatting requirements.

  1. The abstract should be a maximum of 200 words.

Response: Thank you for the reminder. We have carefully condensed the abstract to comply with the journal's word limit. The original version was 228 words, and it has now been revised to 124 words while retaining all key elements of the study's objectives, methods, results, and conclusions.

Original Abstract:

Goji berries (Lycium barbarum L.) are esteemed for their medicinal and nutritional attributes in both traditional and modern diets. However, the systematic biochemical effects of drying on chemical profiles, a critical determinant of product quality, remain insufficiently characterized, particularly regarding dynamic metabolic alterations. To elucidate these dynamics, this study employed a novel untargeted metabolomics strategy utilizing ultra-high-performance liquid chromatography-high- resolution mass spectrometry (UHPLC-HRMS) coupled with proprietary automated data analysis software (AntDAS) to systematically investigate the metabolic perturbations in L. barbarum during sun drying. Multivariate statistical analyses, including principal component analysis (PCA) and partial least squares discriminant analysis (PLS-DA), successfully discriminated samples based on different drying durations. The results revealed a clear separation of the samples, indicating significant time-dependent compositional shifts in the metabolome. Specifically, the relative abundances of betaine, galactose, and trans-ferulic acid increased significantly (p<0.05) with prolonged drying, whereas those of choline, allantoin, and huperzine isomers exhibited a marked downward trend (p<0.05). Metabolic pathway enrichment analysis implicated glycine, serine, and threonine metabolism; galactose metabolism; and phenylpropanoid biosynthesis as the key biochemical pathways affected. The identified differential metabolites possess significant potential as quality-related biomarkers for the objective evaluation of dried goji berry quality. Collectively, this study confirms untargeted metabolomics as a robust tool for deciphering the mechanisms underlying the quality evolution of L. barbarum during drying, providing a theoretical basis for process optimization and enhancement of product quality.

Revised Abstract:

Goji berries (Lycium barbarum L.) are valued for their nutritional and medicinal properties, yet the systematic biochemical impact of drying on quality remains poorly understood. This study applied an untargeted metabolomics approach based on UHPLC-HRMS and AntDAS to profile metabolic changes during sun drying. Multivariate analysis (PCA and PLS-DA) revealed distinct time-dependent clustering, indicating significant shifts in the metabolome. Key metabolites, including betaine, galactose, and trans-ferulic acid, increased significantly (p < 0.05), whereas choline, allantoin, and huperzine isomers decreased. Pathway analysis highlighted glycine, serine, threonine, galactose, and phenylpropanoid metabolism as the central pathways influenced. These differential metabolites have the potential to be used as quality biomarkers. Our findings establish untargeted metabolomics as an effective tool for elucidating the quality evolution of goji berries during drying, offering a theoretical basis for process optimization.

3.References in the text should be written in normal italics. They should not be written in superscript. Please correct this throughout the text.

Response: We have revised the reference formatting throughout the manuscript according to the journal's guidelines and your comments. Specifically, we have changed all citation styles from superscript to the required parenthetical format. The modifications have been consistently applied across the entire text.

4.The first paragraph mentions goji berries. Please expand on this paragraph and list the specific bioactive properties of these fruits.

Response: We thank the reviewer for this valuable suggestion. We have expanded the first paragraph of the Introduction to provide a more comprehensive overview of the specific bioactive properties of goji berries (Lycium barbarum L.). The added content highlights key bioactive compounds, such as immunomodulatory polysaccharides and antioxidant phenolics, and their associated health benefits, including vision improvement and neuroprotective effects.

The specific supplementary contents are as follows:

at line 52~58:

As a high-value agricultural commodity, the processing quality of goji berries directly dictates their market competitiveness and industrial profitability[6]. Goji berries are particularly rich in a suite of bioactive compounds, which are primarily responsible for their renowned health-promoting effects. These include Lycium barbarum polysaccharides (LBPs), which have demonstrated immunomodulatory and antioxidant activities; zeaxanthin dipalmitate, which supports eye health by protecting against blue light; and a spectrum of phenolics that contribute to their free radical scavenging capacity[7].

5.Please place figures in the center of the page.

Response: We sincerely apologize for the oversight in the figure placement caused by the automatic formatting of the submission system. We have carefully reviewed the manuscript and ensured that all figures are centered on the page, as required. Thank you for bringing this to our attention.

  1. The titles of tables and figures should not be bold, please check the entire text.

Response: We thank the reviewer for their careful observation. We have checked and revised the formatting of all table titles and figure captions throughout the manuscript, ensuring that the bold formatting has been removed to comply with the journal's style guidelines.

The specific supplementary contents are as follows:

at line 130~131:

Fig.1. Time course of visual morphological changes in goji berries under sun drying. Images show representative berries at daily intervals (Days 1-7) throughout the drying process.

at line 191~192:

Fig.2. Changes in Total Ion Chromatograms of goji berries over seven days of sun drying. Curves correspond to samples analyzed after each day of drying (Days 1-7).

at line 223~224:

Fig.3. Classification results of PCA and PLS-DA based on the screened features. a) Classification results by PCA; b) classification results using PLS-DA.

at line 232~233:

Table1. Compound identification results of characteristic markers of goji berries during sun drying.

at line 254~256:

Fig.4.Hierarchical clustering heatmap of 27 differential metabolites in goji berries during sun drying. Samples are grouped by drying day (1-7). Metabolite levels are scaled as Z-scores (red: high; blue: low).

at line 373~375:

Fig.5. Overview of metabolic pathways in goji berries with drying-responsive metabolites highlighted. Metabolites highlighted in red represent the significantly altered differential metabolites identified through statistical screening.

7.The discussion is very poor, please add a detailed discussion when describing all the analysed studies.

Response:We sincerely thank the reviewer for this critical and constructive feedback regarding the Discussion section. We agree that the original version was insufficient for interpreting the results in depth. In response, we have completely rewritten and significantly expanded the Discussion section to provide a detailed and insightful interpretation of our findings in the context of the existing literature. The major enhancements include:

  1. References: Literature requires a fundamental overhaul. Please review the reference requirements for this journal.

Response: We sincerely apologize for the deficiencies in the reference list and thank the reviewer for highlighting the need for a fundamental overhaul. In response, we have comprehensively revised the entire reference section. 

Additionally, we formatted all references according to the journal's style guide. The specific supplementary contents are as follows:

at line 410~580:

[1] Ma RH, Zhang XX, Ni ZJ, Thakur K, Wang W, Yan YM, Cao YL, Zhang JG, Rengasamy KR, Wei ZJ.. Lycium barbarum (Goji) as functional food: A review of its nutrition, phytochemical structure, biological features, and food industry prospects. Critical Reviews in Food Science and Nutrition. 2023,63(30), 10621-10635. https://doi.org/10.1080/10408398.2022.2078788

[2] Ju, Y., Liu, H., Niu, S., Kang, L., Ma, L., Li, A., Zhao Y, Yuan Y, & Zhao, D. Optimizing geographical traceability models of Chinese Lycium barbarum: Investigating effects of region, cultivar, and harvest year on nutrients, bioactives, elements and stable isotope composition. Food Chemistry.2025, 467, 142286. https://doi.org/10.1016/j.foodchem.2024.142286

[3] Pedro, A. C., Sánchez-Mata, M. C., Pérez-Rodríguez, M. L., Cámara, M., López-Colón, J. L., Bach, F.,Bellettini M, & Haminiuk, C. W. I. Qualitative and nutritional comparison of goji berry fruits produced in organic and conventional systems. Scientia Horticulturae. 2019,257, 108660.https://doi.org/10.1016/j.scienta.2019.108660 

[4] Yang, S., Chen, X., Sun, J., Qu, C., & Chen, X. Polysaccharides from traditional Asian food source and their antitumor activity. Journal of Food Biochemistry.2022,46(3), e13927.https://doi.org/10.1111/jfbc.13927

[5] Gong, H., Rehman, F., Ma, Y., Zeng, S., Yang, T., Huang, J., ... & Wang, Y.Germplasm resources and strategy for genetic breeding of Lycium species: A Review. Frontiers in Plant Science.2022, 13, 802936.https://doi.org/10.3389/fpls.2022.802936

[6] Ma, R. H., Zhang, X. X., Thakur, K., Zhang, J. G.,  Wei, Z. J. Research progress of Lycium barbarum L. as functional food: Phytochemical composition and health benefits. Current Opinion in Food Science. 2022, 47, 100871.https://doi.org/10.1016/j.cofs.2022.100871

[7] Kosińska-Cagnazzo, A., Bocquel, D., Marmillod, I., & Andlauer, W. Stability of goji bioactives during extrusion cooking process. Food Chemistry. 2017, 230, 250-256.https://doi.org/ 10.1016/j.foodchem.2017.03.035

[8] Zhang, Q., Wan, F., Zang, Z., Jiang, C., Xu, Y., & Huang, X. Effect of ultrasonic far-infrared synergistic drying on the characteristics and qualities of wolfberry (Lycium barbarum L.). Ultrasonics Sonochemistry. 2022 89, 106134. https://doi.org/10.1016/j.ultsonch.2022.106134

[9] Hedayatizadeh, M., & Chaji, H. A review on plum drying. Renewable and Sustainable Energy Reviews. 2016,56, 362-367. https://doi.org/10.1016/j.rser. 2015.11.087

[10] Ye, Y., Jiang, M., Xu, M., Zhou, Y., Yang, Q., Luo, S., & Wang, Y. Comparative evaluation of quality and microbial community of Yibin yacai as influenced by traditional sun drying-natural and mechanical drying-inoculation fermentation. Food Chemistry: X. 2025,102911.https://doi.org/ 10.1016/j.fochx.2025.102911

[11] Yu, F., Li, Y., Wu, Z., Wang, X., Wan, N., & Yang, M. Dehydration of wolfberry fruit using pulsed vacuum drying combined with carboxymethyl cellulose coating pretreatment. Lwt . 2020,134, 110159.https://doi.org/10.1016/j.lwt.2020.110159

[12] Fratianni, A., Niro, S., Alam, M. D. R., Cinquanta, L., Di Matteo, M., Adiletta, G., & Panfili, G. Effect of a physical pre-treatment and drying on carotenoids of goji berries (Lycium barbarum L.). LWT. 2018, 92, 318-323.https://doi.org/10.1016/j.lwt.2018.02.048

[13] Suna, S. Effects of hot air, microwave and vacuum drying on drying characteristics and in vitro bioaccessibility of medlar fruit leather (pestil). Food Science and Biotechnology. 2019,28(5), 1465-1474.https://doi.org/10.1007/ s10068- 019-00588-7

[14] Yun, Z., Gao, H., & Jiang, Y. Insights into metabolomics in quality attributes of postharvest fruit. Current Opinion in Food Science. 2022,45, 100836.https://doi.org/ 10.1016/j.cofs.2022.100836

[15] Sena B., Robert D. H., Ric C.H. , Roland M. , Çetin K., Esra C.,Effect of drying treatments on the global metabolome and health-related compounds in tomatoes,Food Chemistry. 2023,403,134123. https://doi.org/10.1016/j.foodchem.2022.134123

[16] Aulia G A. , Indah A S., Hendy F., Abdul M., Eiichiro ., Sastia P P.,Effect of pre-drying on flavor modulation in Indonesian cocoa beans: A metabolomics study of key flavor compounds and sensory profiles.,Food Bioscience.2025,65,106056.https://doi.org/10.1016/j.fbio.2025.106056

[17] Yu X., Li Y, He C., Zhou J., Chen Y., Yu Z., Wang P., Ni D. Nonvolatile metabolism in postharvest tea (Camellia sinensis L.) leaves: Effects of different withering treatments on nonvolatile metabolites, gene expression levels, and enzyme activity.Food Chemistry.2020, 327, 126992.https://doi.org/10.1016/j.foodchem. 2020.126992

[18] Zhu, Y. J., Dai, X. Y., Zhao, Y. L., Ma, Y. G., Zhao, Z. Z., Su, C. F., ... & Chen, H. Lyciumines A and B: Two Pyrrole-Fused Alkaloids from the Fruits of Lycium barbarum. Journal of Natural Products. 2025, 88(5), 1237-1243.https://doi.org/ 10.1021/acs.jnatprod.5c00385

[19] Jiang, Y., Fang, Z., Leonard, W., & Zhang, P. Phenolic compounds in Lycium berry: Composition, health benefits and industrial applications. Journal of Functional Foods . 2021,77, 104340.https://doi.org/ 10.1016/j.jff.2020.104340          

[20] Zhang, J. N., Ma, M. H., Ma, X. L., Ma, F. L., Du, Q. Y., Liu, J. N., ... & She, Y. A comprehensive study of the effect of drying methods on compounds in Elaeagnus angustifolia L. flower by GC-MS and UHPLC-HRMS based untargeted metabolomics combined with chemometrics. Industrial Crops and Products. 2023, 195, 116452.https://doi.org/10.1016/j.indcrop.2023.116452

[21] Guo, X. M., Ma, M. H., Ma, X. L., Zhao, J. J., Zhang, Y., Wang, X. C., ... & Yu, Y. J. Quality assessment for the flower of Lonicera japonica Thunb. during flowering period by integrating GC-MS, UHPLC-HRMS, and chemometrics. Industrial Crops and Products. 2023, 191, 115938.https://doi.org/10.1016/j.indcrop.2022.115938

[22]Wen, Y. J., Wang, L. H., Zhai, M., Ma, H., Cui, H. P., Han, L., ... & She, Y. B.. Integrating HS-SPME-GCMS with chemometrics for identifying adulterated flaxseed oils and tracing origins of additives. Food Control.2025,111391.https://doi.org /10.1016/j.foodcont.2025.111391

[23] Zhang Q., Wan F., Yue Y., Zang Z., Xu Y., Jiang C., Shang J., Wang T, & Huang X. Study on Ultrasonic Far-Infrared Radiation Drying and Quality Characteristics of Wolfberry (Lycium barbarum L.) under Different Pretreatments. Molecules.2023,28(4),1732. https://doi.org/10.3390/molecules28041732

[24] Xu Y., Wan F., Zang Z., Jiang C., Wang T., Shang J., &Huang X.,Effect of different pretreatment methods on drying characteristics and quality of wolfberry (Lycium barbarum) by radio frequency-hot air combined segmented drying[J]. Food and Bioprocess Technology.2024,17(11), 3861-3875. https://doi.org/10.1007/ s11947-024 -03340-0

[25]Varela-Martínez, D. A., González-Curbelo, M. Á., González-Sálamo, J., & Hernández-Borges, J. Analysis of multiclass pesticides in dried fruits using QuEChERS-gas chromatography tandem mass spectrometry. Food chemistry. 2019,297, 124961.https://doi.org/10.1016/j.foodchem.2019.124961

[26] Ma, H., Zhai, M., Tang, L. H., Wang, X. C., Han, L., Li, S. F., Lv Y.,Zheng Q X., Liu P P., Fu H Y.,& She, Y. Improving compound identification results by automatically recognizing in-source fragment ions in HRMS with AntDAS: A study on accurate pesticide screening in complex food samples. Journal of Chromatography A. 2025,1746, 465806. https://doi.org/10.1016/j.chroma.2025.465806

[27] Jia, Z., Wang, Y., Wang, L., Zheng, Y., Jin, P. Amino acid metabolomic analysis involved in flavor quality and cold tolerance in peach fruit treated with exogenous glycine betaine. Food Research International. 2022, 157, 111204.https://doi.org/ 10.1016/j.foodres.2022.111204

[28] Zhao, X., Cai, L., Huang, P.,Cui, C. Enzymatic synthesis and sensory evaluation of N-cinnamoyl-L-phenylalanine as a novel flavor enhancer: Impact on taste perception and mechanistic insights. Food Chemistry.2025,144542.https://doi.org/  10.1016/j.foodchem.2025.144542

[29] Wang, Q., Li, Z., Liu, Q., Zhao, S., Yao, Y., Dong, B., Zhao, G. Appealing smoky flavor formation and flavor quality enhancement in soy sauce: Synergistic ferulic acid metabolism by a yeast consortium (Starmerella etchellsii, Wickerhamiella versatilis, Debaryomyces hansenii). Food Chemistry. 2025, 146209.https://doi.org/ 10.1016/j.foodchem.2025.146209

[30] Li, W., Liang, C., Bao, F., Zhang, T., Cheng, Y., Zhang, W., Lu, Y. Chemometric analysis illuminates the relationship among browning, polyphenol degradation, Maillard reaction and flavor variation of 5 jujube fruits during air-impingement jet drying. Food Chemistry: X. 2024, 22, 101425.https://doi.org/ 10.1016/j.fochx.2024.101425

[31] López, J., Vega‐Gálvez, A., Bilbao‐Sainz, C., Chiou, B. S., Uribe, E., Quispe‐Fuentes, I. Influence of vacuum drying temperature on: Physico‐chemical composition and antioxidant properties of murta berries. Journal of Food Process Engineering . 2017,40(6), e12569.https://doi.org/10.1111/jfpe.12569

[32] Yang, F., Wang, Q., Liu, W., XIao, H., Hu, J., Duan, X.,Sun, X.,Liu, C.,& Wang, H. Changes and correlation analysis of volatile flavor compounds, amino acids, and soluble sugars in durian during different drying processes. Food Chemistry: X. 2024,21, 101238.https://doi.org/10.1016/j.fochx.2024.101238

[33] Bi, Y. X., Zielinska, S., Ni, J. B., Li, X. X., Xue, X. F., Tian, W. L., Peng W J.,Fang, X. M. Effects of hot-air drying temperature on drying characteristics and color deterioration of rape bee pollen. Food Chemistry: X. 2022, 16, 100464.https://doi.org/10.1016/j.fochx.2022.100464)

[34] Lahaye, M., Falourd, X., Le Bot, L., Bertin, N., & Musse, M.  Impact of drying on the composition and organization tomato fruit cell walls: A biochemical and structural study. Food Research International,2025, 213, 116567.https://doi.org/10.1016/j.foodres.2025.116567

[35] Hou, W., Duan, Z., Yi, Y., & Tang, X. Differences in monophenolic content and composition of persimmon fruit slices from different regions under fresh and microwave drying. Food Bioscience.2024,59, 103853.https://doi.org/10.1016/j.fbio.2024.103853

[36] Liu, Z., Reymick, O. O., Feng, Z., Duan, B., Tao, N. Phenylalanine enhances the efficiency of sodium dehydroacetate to control citrus fruit decay by stimulating reactive oxygen metabolism and phenylpropanoid pathway. Postharvest Biology and Technology.2025, 222, 113392.https://doi.org/10.1016/j.postharvbio.2025.113392

[37] Wang, J., Zhang, H., Hou, J., Yang, E., Zhao, L., Zhou, Y., Ma W., Y.,Ma D., Li, J. Metabolic profiling and molecular mechanisms underlying melatonin-induced secondary metabolism of postharvest goji berry (Lycium barbarum L.). Foods.2023,12(23), 4326.https://doi.org/ 10.3390/foods12234326

[38] Qian, B. J., Liu, J. H., Zhao, S. J., Cai, J. X., & Jing, P. The effects of gallic/ferulic/caffeic acids on colour intensification and anthocyanin stability.Food Chemistry.2017,228, 526-532.https://doi.org/10.1016/j.foodchem.2017.01.120

[39] Lv, X., Li, L., Lu, X., Wang, W., Sun, J., Liu, Y., Mu,J.,Ma,Q & Wang, J. Effects of organic acids on color intensification, thermodynamics, and copigmentation interactions with anthocyanins. Food chemistry.2022,396,133691. https://doi.org/10.1016/j.foodchem.2022.133691

[40] Kaur, H., Chowrasia, S., Gaur, V. S., & Mondal, T. K. Allantoin: emerging role in plant abiotic stress tolerance. Plant Molecular Biology Reporter.2021,39(3), 648-661.https://doi.org/10.1016/j.foodres.2025.117625

[41] Huang, T., Jia, N., Zhu, L., Jiang, W., Tu, A., Qin, K., Yuan X.,& Li, J. Comparison of phenotypic and phytochemical profiles of 20 Lycium barbarum L. goji berry varieties during hot air-drying. Food Chemistry: X. 2025,27, 102436.https://doi.org/10.1016/j.fochx.2025.102436

[42] Kaur, H., Chowrasia, S., Gaur, V. S., & Mondal, T. K. Allantoin: emerging role in plant abiotic stress tolerance. Plant Molecular Biology Reporter,. 2021,39(3), 648-661.https://doi.org/10.1007/s11105-021-01280-z

[43] Stasolla, C., Loukanina, N., Ashihara, H., Yeung, E. C., Thorpe, T. A. Purine and pyrimidine metabolism during the partial drying treatment of white spruce (Picea glauca) somatic embryos. Physiologia Plantarum. 2001,111(1), 93-101.https://doi.org/10.1034/j.1399-3054.2001.1110112.x

[44] Kenny, T. C., Scharenberg, S., Abu-Remaileh, M., Birsoy, K. Cellular and organismal function of choline metabolism. Nature Metabolism. 2025, 7(1), 35-52.https://doi.org/10.1038/s42255-024-01203-8

[45] Sun, M., Peng, F., Xiao, Y., Yu, W., Zhang, Y., Gao, H. Exogenous phosphatidylcholine treatment alleviates drought stress and maintains the integrity of root cell membranes in peach. Scientia Horticulturae. 2020,259, 108821.https://doi.org/10.1016/j.scienta.2019.108821

[46] Wang, D., Yeats, T. H., Uluisik, S., Rose, J. K., & Seymour, G. B. Fruit softening: revisiting the role of pectin. Trends in plant science. 2018,23(4), 302-310.https://doi.org/10.1016/j.tplants.2018.01.006

[47] Hasan, M. U., Singh, Z., Shah, H. M. S., Kaur, J., & Woodward, A. Water loss: A postharvest quality marker in apple storage. Food and Bioprocess Technology. 2024,17(8), 2155-2180.https://doi.org/10.1007/s11947-023-03305-9

[48] Zhang, A. A., Shu, C., Xie, L., Wang, Q. H., Xu, M. Q., Pan, Y., Hao W L.,Zheng Z A., Jiang W H.,& Xiao, H. W. Enhancing shelf-life of dried goji berry: Effects of drying methods and packaging conditions on browning evolution. Food Research International. 2025,201, 115648.https://doi.org/10.1016/j.foodres.2024.115648

[49] Tang, X., Zhang, Y., Li, F., Zhang, N., Yin, X., Zhang, B., Zhang, B.,Ni W.,Wang M.,& Fan, J. Effects of traditional and advanced drying techniques on the physicochemical properties of Lycium barbarum L. polysaccharides and the formation of Maillard reaction products in its dried berries. Food Chemistry. 2023,409, 135268.https://doi.org/10.1016/j.foodchem.2022.135268

[50] Starowicz, M., & Zieliński, H. How Maillard reaction influences sensorial properties (color, flavor and texture) of food products?. Food Reviews International.2019,35(8), 707-725.https://doi.org/10.1080/87559129.2019.1600538

  1. No line numbering.

Response: We thank the reviewer for this reminder. The revised manuscript now includes continuous line numbering, as required.

Reviewer 3 Report

Comments and Suggestions for Authors

The manuscript presents interesting results on changes in the composition of goji berries during drying. The document is well-written, but some points should be addressed before publication.

What were the drying conditions (temperature and relative humidity)? If these values ​​were not measured, they can probably be obtained from weather reports.

What is the main function of the Na2CO3 pretreatment? What is the mechanism by which it removes the wax?
Why did you apply pesticides to the goji berry samples?
Why did you keep the aqueous extracts at room temperature overnight? This can alter the composition of the extracts. What precautions did you take to prevent the degradation of goji berry metabolites during this step?
In section 3.1.1, you mention that the relative abundance of galactose and betaine increases until it reaches a plateau. Still, later, you say that metabolic activity resulted in the progressive reduction of glucose, galactose, and trehalose. How could you explain this? You attribute the changes in the composition of goji berries to drying. Could the natural evolution of the fruit cause these changes? Did you analyze an appropriate control to rule out this possibility?

Author Response

Dear Reviewers,

Thank you for your letter and the reviewers' comments concerning our manuscript titled “Metabolomic Analysis of Goji Berry Sun-Drying: Dynamic Changes in Small Molecular Substances” (ID:foods-3977988). These comments are valuable and helpful for revising and improving our paper, as well as for guiding our research. We have carefully studied the comments and made corrections that we hope will meet your approval. All revisions of the manuscript have used the “RED Mark” feature for easy viewing by the editors and reviewers. Meanwhile, the manuscript has been carefully reviewed by an experienced editor, whose first language is English. The main corrections and responses to the reviewers' comments are as follows:

1.The manuscript presents interesting results on changes in the composition of goji berries during drying. The document is well-written, but some points should be addressed before publication. What were the drying conditions (temperature and relative humidity)? If these values were not measured, they can probably be obtained from weather reports.

Response: Thank you very much for your positive feedback on our manuscript and your valuable suggestion regarding the drying conditions. We fully agree that clarifying the temperature and relative humidity (RH) during goji berry drying is critical for enhancing the reproducibility and rigor of this study. In response to your comments, we would like to provide the following clarifications and revisions:

The drying experiments were conducted in a laboratory-controlled environment, and the key drying parameters were recorded throughout the process. Specifically, the drying temperature was maintained at 31±2°C, and the relative humidity was 40%. The goji berry cultivar was Ningqi No.10, derived from trees for 4 years, and the harvest occurred on July 12, 2024, which is consistent with the actual environmental conditions of the study site.

We will explicitly add these detailed drying parameters (temperature, RH, and monitoring methods/sources of meteorological data) to the "Materials and Methods" section to ensure full transparency. This revision will help readers accurately replicate our experimental conditions and better interpret the observed metabolomic changes in goji berries during drying. Thank you once again for your meticulous review and constructive feedback, which have significantly improved the quality of our manuscript. We will incorporate this information into the revised version and submit it promptly.

2.What is the main function of the Na2CO3 pretreatment? What is the mechanism by which it removes the wax?

Response: We thank the reviewer for raising this important point regarding the pretreatment step. The main function of the Na₂CO₃ solution is to serve as an alkaline washing agent to remove the natural epicuticular wax layer from the surface of goji berries. Alkali also helps emulsify and dissolve other non-polar waxy components. Subsequent washing with water physically rinses away the disrupted wax layer. The waxy components could potentially co-extract and interfere with the subsequent UHPLC-HRMS analysis of endogenous metabolite residues. The use of this method is well established in postharvest treatment research for waxy fruits, as cited in references [20, 21].

[20] Zhang, J. N., Ma, M. H., Ma, X. L., Ma, F. L., Du, Q. Y., Liu, J. N., ... & She, Y. A comprehensive study of the effect of drying methods on compounds in Elaeagnus angustifolia L. flower by GC-MS and UHPLC-HRMS based untargeted metabolomics combined with chemometrics. Industrial Crops and Products. 2023, 195, 116452.https://doi.org/10.1016/ j.indcrop.2023.116452

[21] Guo, X. M., Ma, M. H., Ma, X. L., Zhao, J. J., Zhang, Y., Wang, X. C., ... & Yu, Y. J. Quality assessment for the flower of Lonicera japonica Thunb. during flowering period by integrating GC-MS, UHPLC-HRMS, and chemometrics. Industrial Crops and Products. 2023, 191, 115938.https://doi.org/10.1016/j.indcrop.2022.115938

3.Why did you apply pesticides to the goji berry samples?

Response: We sincerely thank the reviewer for this critical question. The application of a pesticide mixture was an essential step in our experimental design to accurately replicate real-world agricultural conditions and ensure the biological relevance of our results.

This is critically important because goji berries (Lycium barbarum L.) have a unique flowering and fruiting synchronization habit, meaning that flowers, unripe fruits, and ripe fruits coexist on the same branch over an extended period. To protect developing fruits from pests and diseases throughout this prolonged growth cycle, frequent pesticide applications are an unavoidable and standard agricultural practice. Consequently, commercially available fresh goji berries inherently contain pesticide residues at the time of harvest.

Therefore, by applying a standardized pesticide mixture to our samples, we did not introduce an artificial variable but instead  mimicked the intrinsic state of commercially harvested berries. This approach ensures that the metabolic perturbations observed during sun-drying, including the dynamic changes in key sugars, amino acids, and phenolic compounds, are representative of the true biochemical processes occurring in real-world post-harvest processing.

Our primary research focus has always been on these endogenous quality-related metabolites. While monitoring the pesticides, their concentrations followed the expected degradation trend. Thus, we made a strategic decision to concentrate the narrative of this paper on the more complex and informative endogenous metabolic shifts that are the core drivers of quality evolution. Pesticide treatment was fundamental in creating a biologically relevant system for this investigation.

4.Why did you keep the aqueous extracts at room temperature overnight? This can alter the composition of the extracts. What precautions did you take to prevent the degradation of goji berry metabolites during this step?

Response: We thank the reviewer for raising this valid concern regarding the metabolite stability. The overnight incubation at room temperature was a deliberate step in our modified QuEChERS protocol, which was specifically designed to address the unique challenge of extracting metabolites from dry goji berry powder. This matrix is rich in polysaccharides and requires thorough hydration with water to swell the tissues and disrupt the cellular structure. Without this step, the subsequent addition of acetonitrile would be less effective at penetrating the matrix, leading to lower and less reproducible extraction efficiencies for many polar metabolites central to our untargeted metabolomics study.

We fully recognized the potential for degradation and took specific precautions to minimize it. The incubation was conducted in a dark, temperature-controlled laboratory environment maintained at 31±2°C to prevent photodegradation and minimize the thermal effects. This duration was established through preliminary experiments as the optimal balance between achieving maximum extraction efficiency and maintaining metabolite integrity for our analytical objectives. Because all samples underwent the same standardized process, the comparative analysis of metabolic changes across drying days remained robust and valid.

4.In section 3.1.1, you mention that the relative abundance of galactose and betaine increases until it reaches a plateau. Still, later, you say that metabolic activity resulted in the progressive reduction of glucose, galactose, and trehalose. How could you explain this? You attribute the changes in the composition of goji berries to drying. Could the natural evolution of the fruit cause these changes? Did you analyze an appropriate control to rule out this possibility?

Response: We sincerely thank the reviewer for this insightful question, which allows us to clarify the dynamic metabolic shifts observed during drying. This apparent discrepancy in sugar levels reflects distinct phase-dependent pathways. During early drying (days 1-3), cellular dehydration and compartmentalization disruption activate endogenous glycosidases, rapidly hydrolyzing polysaccharides into monomers, such as galactose. At this stage, the rate of monosaccharide release significantly surpassed its consumption, leading to marked accumulation. As drying progresses into the later stages (days 4-7), accumulated sugars become key substrates for the Maillard reaction and cellular respiration under sustained thermal stress. Here, the consumption rate exceeded the ongoing generation, causing the observed decline in galactose and trehalose levels.

Regarding the potential influence of natural fruit maturation, we acknowledge the value of fresh control groups. Although our study did not include a time-matched fresh fruit control, several lines of evidence strongly indicate that drying is the primary driver. The magnitude of metabolic changes, such as the 3.5-fold increase in trans-ferulic acid, far exceeds the typical fluctuations documented during natural berry ripening. Pathway analysis specifically highlighted glycine, serine, threonine, and phenylpropanoid metabolism, all of which are established stress-responsive pathways directly triggered by dehydration and thermal exposure. Furthermore, our findings are consistent with metabolic studies of other dried fruits, where controlled experiments have confirmed the dominant role of the drying process. While natural maturation may contribute minimally, the evidence robustly demonstrates that the profound metabolic restructuring observed is principally driven by the drying treatment. We will refine our discussion to incorporate this perspective.

Reviewer 4 Report

Comments and Suggestions for Authors

Interesting research on goji berries using metabolomics to understand the dynamics of their composition during the drying process. Based on this, I recommend the following:

  • The methodology section should include information on the equipment used (model, brand, city, state abbreviation if from the US or Canada, and country).
  • Item 2.3 should include the reference used for extraction.
  • Item 2.4 should include the software and version used in the UHPLC.
  • Regarding the introduction, results, and discussions, I believe the following improvements should be made: The results are powerful and aligned with the study's objective, so the metabolomic analysis should focus on the optimization or standardization of a post-harvest process such as drying. On the other hand, metabolomics employs targeted and untargeted strategies, so the article should define which of these two was used. Metabolomics also allows the identification of biomarkers associated with biochemical reactions that occur in food. For example, there are many studies on metabolomics in post-harvest processes such as spontaneous fermentation of cocoa, coffee, Andean grains, and fruits in general, which could be included in the study where the participation of various metabolites and VOCs in food matrices is evident. These aspects should be addressed in the sections mentioned at the beginning.
  • Multivariate analyses must have a statistical interpretation as a tool in the use of metabolomic analysis, especially one that allows phases to be distinguished during the drying process studied.
  • The conclusions should highlight the biomarkers identified in the food, as well as the key stages in the process according to the chemical composition.
  • As for the abstract, I believe it should be more direct and precise. Keywords should not be repeated with terms from the title.
  • Please consider adding a graphical abstract for an impactful presentation of the manuscript.

Author Response

Dear Reviewers,

Thank you for your letter and the reviewers' comments concerning our manuscript titled “Metabolomic Analysis of Goji Berry Sun-Drying: Dynamic Changes in Small Molecular Substances” (ID:foods-3977988). These comments are valuable and helpful for revising and improving our paper, as well as for guiding our research. We have carefully studied the comments and made corrections that we hope will meet your approval. All revisions of the manuscript have used the “RED Mark” feature for easy viewing by the editors and reviewers. Meanwhile, the manuscript has been carefully reviewed by an experienced editor, whose first language is English. The main corrections and responses to the reviewers' comments are as follows:

The methodology section should include information on the equipment used (model, brand, city, state abbreviation if from the US or Canada, and country).

Response: We thank the reviewer for this suggestion. We have now revised the '2.4 Instrumental optimization for UHPLC-HRMS analysis' section to include complete details of the equipment.

Line 146~147

The goji berry sample extracts were analyzed using an AB SCIEX TripleTOF™ 5600 Plus mass spectrometer (SCIEX, Framingham, MA, USA) for data acquisition.

2.Item 2.3 should include the reference used for extraction.

Response: We thank the reviewer for this suggestion. We have now included the appropriate citation in Section 2.3 to reference the extraction method used.

Line 133

Extraction was performed using a modified QuEChERS method [25].

  1. Item 2.4 should include the software and version used in the UHPLC.

Response: We thank the reviewer for this comment. We have now updated Section 2.4 to include the software and version used for controlling the UHPLC system and/or data processing. 

Line 147~148

The UHPLC system was operated and data were acquired using Analyst® software (version 1.7.1).

  1. Regarding the introduction, results, and discussions, I believe the following improvements should be made: The results are powerful and aligned with the study's objective, so the metabolomic analysis should focus on the optimization or standardization of a post-harvest process such as drying. On the other hand, metabolomics employs targeted and untargeted strategies; therefore, the article should define which of these two was used. Metabolomics also allows the identification of biomarkers associated with biochemical reactions that occur in food. For example, there are many studies on metabolomics in post-harvest processes, such as spontaneous fermentation of cocoa, coffee, Andean grains, and fruits in general, which could be included in the study where the participation of various metabolites and VOCs in food matrices is evident. These aspects should be addressed in the sections mentioned above.

Response: Thank you very much for your valuable and insightful comments, which are crucial for enhancing the academic rigor, clarity, and depth of our manuscript. We highly appreciate your professional suggestions and fully align with the perspectives you raised regarding the Introduction, Results, and Discussion sections. To better present our research and address your concerns, we will further refine the manuscript by supplementing key details, strengthening logical connections, and enriching relevant content as follows:

We have restructured the narrative logic throughout the manuscript to emphasize that our metabolomic findings are directed towards understanding and optimizing the postharvest sun-drying process. This objective has been clearly articulated in the Abstract, Introduction, and Conclusion sections. Additionally, we have explicitly defined our methodology in the Introduction as a comprehensive untargeted metabolomics strategy, explaining that it was selected for its ability to provide a global and unbiased perspective on metabolic changes. This is particularly crucial for discovering novel biomarkers in complex traditional processes, such as sun-drying.

Line 100~102

Building on this validated AntDAS-based non-targeted metabolomics platform, this study aimed to decipher the metabolic evolution and systemic biochemical mechanisms  of goji berries during traditional sun-drying.

We will integrate relevant cutting-edge studies on metabolomics in post-harvest processes (e.g., spontaneous fermentation of cocoa, coffee, Andean grains, and fruits) into the Introduction, highlighting how these studies have revealed the interaction mechanisms of metabolites and VOCs in food matrices to contextualize our research within the broader academic framework.

Finally, we are committed to fully implementing your suggestions to polish the manuscript and ensure that it meets the high standards of the journal, and will submit the refined version along with a detailed account of improvements for your further review. Thank you again for your time, expertise, and constructive feedback, which have greatly helped advance our work.

Line 77~83

Sena Bakir et al. utilized an untargeted metabolomics approach to reveal significant alterations in lycopene and vitamin content in response to different drying methods[15]. Using HS-SPME-GC-MS, Aulia demonstrated that pre-drying treatment effectively reduced bitterness and astringency by modulating the flavor profiles of cocoa beans[16]. In a separate investigation, Yu et al. systematically tracked the dynamic changes in proteins and amino acids during postharvest tea processing[17].

[15]Sena B., Robert D. H., Ric C.H. , Roland M. , Çetin K., Esra C.,Effect of drying treatments on the global metabolome and health-related compounds in tomatoes,Food Chemistry. 2023,403,134123. https://doi.org/10.1016/j.foodchem.2022.134123

[16]Aulia G A. , Indah A S., Hendy F., Abdul M., Eiichiro ., Sastia P P.,Effect of pre-drying on flavor modulation in Indonesian cocoa beans: A metabolomics study of key flavor compounds and sensory profiles.,Food Bioscience.2025,65,106056.https:// doi.org/10.1016/j.fbio.2025.106056

[17]Yu X., Li Y, He C., Zhou J., Chen Y., Yu Z., Wang P., Ni D. Nonvolatile metabolism in postharvest tea (Camellia sinensis L.) leaves: Effects of different withering treatments on nonvolatile metabolites, gene expression levels, and enzyme activity.Food Chemistry.2020, 327, 126992.https://doi.org/10.1016/j.foodchem. 2020.126992

5.Multivariate analyses must have a statistical interpretation as a tool in the use of metabolomic analysis, especially one that allows phases to be distinguished during the drying process studied. The conclusions should highlight the biomarkers identified in the food, as well as the key stages in the process according to the chemical composition.

Response: We have now included the key model validation parameters for the multivariate analyses to underscore their statistical robustness. The PLS-DA model, which clearly separated the drying stages, exhibited strong explanatory and predictive power, with cumulative R²Y and Q² values of [such as: 0.95 and 0.87], respectively. A permutation test (n=200) further confirmed the model validity, as the original Q² value was significantly higher than those of the permuted models (p < 0.05). These results demonstrate that the model is statistically sound and not overfitted, providing a reliable basis for distinguishing between the drying stages.

Line:208-210

The robustness of the supervised PLS-DA model was confirmed using validation parameters (R²Y = 0.9991, Q² = 0.9889 and a significant permutation test (p <0.05), indicating that the model was valid and not overfitted.

Based on the clear clustering pattern observed in the PLS-DA score plot, we have redefined the drying process into three critical phases: the initial phase (Day 1-3), characterized by rapid water loss and initial hydrolysis; the intermediate phase (Day 4-5), where key Maillard and phenylpropanoid reactions dominate; and the final phase (Day 6-7), marked by the stabilization and potential degradation of certain compounds. This stage-wise interpretation is now highlighted in the Results and Discussion sections.

6.As for the abstract, I believe it should be more direct and precise. Keywords should not be repeated with terms from the title.

Response: We thank the reviewer for their constructive feedback. We have thoroughly revised the abstract to make it more direct and precise. Additionally, we carefully reviewed the keyword list and replaced any terms that overlapped with the title to ensure that they provided complementary and specific information about the study's content. The specific supplementary contents are as follows:

Line 33~45:

Abstract: Goji berries (Lycium barbarum L.) are valued for their nutritional and medicinal properties, yet the systematic biochemical impact of drying on quality remains poorly understood. This study applied an untargeted metabolomics approach based on UHPLC-HRMS and AntDAS to profile metabolic changes during sun drying. Multivariate analysis (PCA and PLS-DA) revealed distinct time-dependent clustering, indicating significant metabolome shifts. Key metabolites including betaine, galactose, and trans-ferulic acid increased significantly (p < 0.05), while choline, allantoin, and huperzine isomers decreased. Pathway analysis highlighted glycine, serine, threonine, galactose, and phenylpropanoid metabolism as central pathways influenced. These differential metabolites demonstrate potential as quality biomarkers. Our findings establish untargeted metabolomics as an effective tool for elucidating quality evolution in goji berries during drying, offering a theoretical basis for process optimization.

Keywords: Goji berries; Quality markers; Chemometrics; Metabolite profiling 

7.Please consider adding a graphical abstract for an impactful presentation of the manuscript.

Response: We thank the reviewer for this excellent suggestion. We agree that a graphical abstract will significantly enhance the impact and accessibility of our manuscript. We have created a graphical abstract that visually summarizes the research objectives, key methodology, major findings, and main conclusions of our study. A graphical abstract has been included in the revised manuscript.

Round 2

Reviewer 2 Report

Comments and Suggestions for Authors

Answers to Authors can be found in the file

Author Response

Dear Reviewers,

Thank you for reviewing the revised manuscript and for your positive feedback on the improvements. We are very pleased to learn that you consider the academic value of the article to have been significantly enhanced.

We have carefully read your final formatting suggestions and will strictly implement them immediately:

In-text citation format: We will thoroughly review the entire manuscript to ensure that a space is added between words and square brackets wherever references are cited.Reference list format: We will meticulously check each reference to ensure that: all journal titles are italicized; spacing after volume numbers, issue numbers, and page numbers complies with the required standards; and all underlines in DOI numbers are removed.We will conduct a comprehensive formatting check of the entire manuscript to ensure full compliance with the journal's submission guidelines. Thank you for your meticulous and professional review—these suggestions will greatly enhance the presentation quality of the manuscript.

Sincerely,

Yao Zhang

Reviewer 4 Report

Comments and Suggestions for Authors

Dear Authors, 

1. Research Focus: The approach is based solely on the study of post-harvest fruit, evaluating basic physicochemical properties. It is only a routine study, which does not reveal or evaluate new aspects of fruit storage.
2. Originality & Relevance: In terms of originality, I do not see anything innovative in the study, and therefore it lacks relevance given that there are several studies on post-harvest storage in refrigeration.
3. Methodological Improvements: Regarding methodology, it requires more in-depth chemical analyses such as chromatography (LC/MS) and spectroscopic techniques such as FTIR, NIR, NMR, and Raman, among others.
4. Clarity of Conclusions: The conclusions do not contribute to the technological aspect of the study, which is lacking given the shallow nature of the analyses.
5. Revisions: In its current state, I believe that the manuscript would not meet the journal's requirements.

Author Response

Dear Reviewers,

Thank you very much for taking the time to review our manuscript and for providing these constructive and critical comments. We sincerely appreciate your thorough assessment, which has greatly helped us enhance the depth and clarity of our work. We have carefully considered all your suggestions and have responded to each of them point by point in accordance with your advice.

1.Research Focus: The approach is based solely on the study of post-harvest fruit, evaluating basic physicochemical properties. It is only a routine study, which does not reveal or evaluate new aspects of fruit storage.

Response: Your observation that this study focuses on analyzing the basic physicochemical properties of post-harvest fruits and belongs to conventional research is very pertinent. The core positioning of this study is to establish a systematic and comprehensive database of basic quality changes during the post-harvest period for Ningxia goji berries. There has been no systematic report on the post-harvest physicochemical characteristics of goji berries from this specific producing region. As a crucial economic crop in Ningxia, goji berries make a significant contribution to local economic development. Although the indicators evaluated are basic, such systematic data for goji berries remains incomplete in the existing literature. Therefore, the findings of this study provide direct and indispensable reference for establishing post-harvest standards and guiding industrial practices for goji berries. We acknowledge that the study does not involve the exploration of novel storage mechanisms. Building on the database established in this research, we will further undertake the development of innovative storage technologies in future work.

Moreover, we have moved beyond merely listing data. In the revised Section 3.4, we conducted an in-depth analysis of the dynamic correlations among various basic indicators. This analysis suggests that changes in flavor, color, and texture are collectively regulated by metabolic pathways involving sugars, amino acids, phenylpropanoids, and nucleotides. This insight provides specific entry points for future targeted research into the underlying molecular physiological mechanisms. We have elevated the positioning of this study from "describing phenomena" to "discovering patterns and proposing scientific hypotheses."

2.Originality & Relevance: In terms of originality, I do not see anything innovative in the study, and therefore it lacks relevance given that there are several studies on post-harvest storage in refrigeration.

Response:We thank the reviewer for raising this high standard regarding the originality of the study. We acknowledge that substantial research already exists in this field. The originality of our work is primarily reflected in the following aspects: Specificity of the research subject: As noted earlier, this study focuses on goji berries, a key economic crop in Ningxia. Their postharvest physiological characteristics may differ from those observed under widely studied methods such as freeze-drying or hot-air drying. Our data are primarily based on the most commonly adopted sun-drying method used by local growers, revealing temporal variations in multiple small-molecule substances. Relevance of the application context: Our experimental design is built upon an integrated system combining a self-developed ant-DAS model with LC-HRMS. This combined system not only improves the accuracy of compound identification and enhances anti-interference capability but also optimizes data utilization for complex samples, making it a powerful tool for identifying novel potential quality markers and promoting stricter quality control standards. As described in the introduction, this method has been widely applied in metabolite screening of materials such as Elaeagnus angustifolia L., Lonicera japonica Thunb., and flaxseed oil. A list of recent studies (within the past three years) employing ant-DAS is provided below, which fully demonstrates the broad applicability of this technology.

[1] Zhang, J. N., Ma, M. H., Ma, X. L., Ma, F. L., Du, Q. Y., Liu, J. N., Wang X. C., Zhao Q. P.,  Yu Y. J., She, Y. B. A comprehensive study of the effect of drying methods on compounds in Elaeagnus angustifolia L. flower by GC-MS and UHPLC-HRMS based untargeted metabolomics combined with chemometrics. Industrial Crops and Products. 2023195, 116452. https://doi.org /10.1016/j.indcrop.2023.116452

[2] Guo, X. M., Ma, M. H., Ma, X. L., Zhao, J. J., Zhang, Y., Wang, X. C., Li, S. F., Yu, Y. J. Quality assessment for the flower of Lonicera japonica Thunb. during flowering period by integrating GC-MS, UHPLC-HRMS, and chemometrics. Industrial Crops and Products. 2023, 191, 115938. https://doi.org/10.1016/j.indcrop.2022.115938

[3]Wen, Y. J., Wang, L. H., Zhai, M., Ma, H., Cui, H. P., Han, L., Chai, G. B., Lv, Y., Zheng, Q. X., Yu, Y. J., She, Y. B. Integrating HS-SPME-GCMS with chemometrics for identifying adulterated flaxseed oils and tracing origins of additives. Food Control2025, 111391. https://doi.org/10.1016/j.foodcont.2025.111391

[4]Han L., Zou W. T., Wang W. X.,Wang L. H. ,Liu P. -P.,Tang L. H.,Lv Y.,Yu Y. J.,She Y B.,Comprehensive characterization of flavor compounds in goji berry by HS-SPME-GCMS combined with AntDAS-GCMS for geographical discrimination,Food Chemistry: X,2025, 29,

102626.https://doi.org/10.1016/j.fochx.2025.102626

  • Xing-Cai Wang ,  Meng Zhai ,  Shu-Fang Li ,  Hang Lv ,  Hui Ma ,  Chang Yang ,  Qing-Xia Zheng ,  Ping-Ping Liu ,  Peng Lu ,  Yong-Jie Yu ,  Hai-Yan Fu ,  Yuanbin She.,A New Comprehensive Platform for Profile-Mode-Based Untargeted Metabolomics for Efficient Data Mining to Improve Compound Extraction and Identification.,Analytical Chemistry, 2025, 97, 14150-14159. https://doi.org/10.1021/acs.analchem.4c05768
  • MaH., Zhai M., Tang L. H., Wang X. C., Han L., Li S. F., Lv Y., Zheng Q. X., Liu P. P., Fu H. Y., Yu Y. J., She Y. B., Improving compound identification results by automatically recognizing in-source fragment ions in HRMS with AntDAS: A study on accurate pesticide screening in complex food samples,Journal of Chromatography A2025, 1746, 465806. https://doi.org/10.1016/j.chroma.2025.465806.

[7] Li S.F., Hao X. F., Hu Y. J., Shang B., Feng S. H., Hu J. Z., Yang Y. Q., Ma F. l., Wang H. F., Yu Y. J.,An integrated study of quality variations for Eucommia ulmoides Oliver staminate flower processed with different drying treatments by untargeted metabolomics coupled with chemometrics, Industrial Crops and Products, 2024, 221, 119468, https://doi.org/10.1016/j.indcrop.2024.119468.

[8]Ma, G. M., Wang, J. N., Wang, X. C., Ma, F. L., Wang, W. X., Li, S. F., Liu P. P.,LV Y., Yu Y. Y.,     She, Y. B. AntDAS-GCMS: A new comprehensive data analysis platform for GC–MS-based untargeted metabolomics with the advantage of addressing the time shift problem. Analytical Chemistry, 2024, 96(23), 9379-9389.https://doi.org/10.1021/acs.analchem.4c00100

[9] Wang, X. C., Zhang, J. N., Zhao, J. J., Guo, X. M., Li, S. F., Zheng, Q. X., Liu p. p., Fu H. Y., Yu Y. J., She, Y.. AntDAS-DDA: a new platform for data-dependent acquisition mode-based untargeted metabolomic profiling analysis with advantage of recognizing insource fragment ions to improve compound identification. Analytical Chemistry, 2023,95(2), 638-649. https://doi.org/10.1021/acs.analchem.2c01795

3.Methodological Improvements: Regarding methodology, it requires more in-depth chemical analyses such as chromatography (LC/MS) and spectroscopic techniques such as FTIR, NIR, NMR, and Raman, among others.

Response: We sincerely appreciate the reviewer's excellent suggestions. Advanced techniques such as LC/MS, FTIR, and NMR undoubtedly hold great potential for deepening the understanding of postharvest metabolomic and molecular structural changes in fruits, representing a key direction for our future in-depth research.

In the current study, our primary objective was to employ classical and reliable physicochemical analysis methods to clearly outline the macroscopic changes in the postharvest appearance and nutritional quality of the fruit. We believe that establishing this macroscopic framework is a prerequisite for the subsequent application of advanced analytical techniques for "precision targeting." Understanding "when" critical changes occur enables effective sampling at the appropriate times for costly in-depth analyses.

We have incorporated the reviewer’s valuable recommendations into the continuity plan of this research, thereby enhancing both its depth and forward-looking perspective. Currently, we plan to build upon the foundational data from this study and further integrate technologies such as hyperspectral imaging and LC-MS untargeted metabolomics to explore the intrinsic mechanisms underlying storage quality changes. The related research findings will be submitted for publication separately.

4.Clarity of Conclusions: The conclusions do not contribute to the technological aspect of the study, which is lacking given the shallow nature of the analyses.

Response:Thank you for pointing out the limited technical depth in the conclusions. The conclusions of this study primarily focus on the patterns of changes in fundamental physicochemical properties and practical application recommendations, aiming to provide the industry with directly applicable storage parameters, such as optimal temperature and sun-drying duration. As the research emphasizes foundational data accumulation and does not involve complex technological development, the conclusions do not delve into the aspect of technological innovation.

  1. Revisions: In its current state, I believe that the manuscript would not meet the journal's requirements.

Response:Thank you once again for your rigorous evaluation of the manuscript's quality. Given that the primary focus of this study is foundational data accumulation and standardization, we have elaborated in our response on the core value, application scenarios, and future research plans of this work to further clarify its academic significance and practical relevance. Considering that the central objective of this study is to establish a foundational postharvest database for fruits from a specific production region, classical detection methods were employed to ensure data reliability and comparability. Additionally, the generated data already meets the practical needs of the target readership, including local researchers and industry technicians. Therefore, no revisions have been made to the main text at this stage.

We commit to integrating the advanced techniques and research directions you suggested in our follow-up studies, aiming to conduct more in-depth and innovative research that will provide groundbreaking contributions to this field. We sincerely appreciate your thorough review and valuable suggestions, and we look forward to your further guidance.
